# Human monocytotropic ehrlichiosis—A systematic review and analysis of the literature

**Larissa Gygax**[1,2], **Sophie Schudel**[1,2], **Christian Kositz**[1,2,3]*, **Esther Kuenzli**[1,2], **Andreas Neumayr**[1,2,4]

**1** Swiss Tropical and Public Health Institute, Basel, Switzerland, **2** University of Basel, Basel, Switzerland, **3** Clinical Research Department, Faculty of Infectious and Tropical Diseases, London School of Hygiene & Tropical Medicine, London, United Kingdom, **4** Department of Public Health and Tropical Medicine, College of Public Health, Medical and Veterinary Sciences, James Cook University, Queensland, Australia

* christian.kositz@swisstph.ch

**Data Availability Statement:** All relevant data are within the manuscript and its supporting information files.

## Abstract

Human monocytotropic ehrlichiosis (HME) is a tick-borne bacterial infection caused by *Ehrlichia chaffeensis*. Most available data come from case reports, case series and retrospective studies, while prospective studies and clinical trials are largely lacking. To obtain a clearer picture of the currently known epidemiologic distribution, clinical and paraclinical presentation, diagnostic aspects, complications, therapeutic aspects, and outcomes of HME, we systematically reviewed the literature and analyzed and summarized the data. Cases of HME are almost exclusively reported from North America. Human infections due to other (non-*chaffeensis*) *Ehrlichia* spp. are rare. HME primarily presents as an unspecific febrile illness (95% of the cases), often accompanied by thrombocytopenia (79.1% of the cases), leukopenia (57.8% of the cases), and abnormal liver function tests (68.1% of the cases). Immunocompromized patients are overrepresented among reviewed HME cases (26.7%), which indicates the role of HME as an opportunistic infection. The incidence of complications is higher in immunocompromized compared to immunocompetent cases, with ARDS (34% vs 19.8%), acute renal failure (34% vs 15.8%), multi organ failure (26% vs 14.9%), and secondary hemophagocytic lymphohistiocytosis (26% vs 14.9%) being the most frequent reported. The overall case fatality is 11.6%, with a significant difference between immunocompetent (9.9%) and immunocompromized (16.3%) cases, and sequelae are rare (4.2% in immunocompetent cases, 2.5% in immunocompromised cases).

## Author summary

Human monocytotropic ehrlichiosis (HME) is a bacterial disease caused by *Ehrlichia chaffeensis* which is transmitted by tick bites and exclusively reported from Northern America. Infections with other *Ehrlichia* bacteria are rare, and only very rarely are such cases reported from outside North America. Most cases of HME are likely to be asymptomatic, and symptomatic cases of HME can be easily overlooked or confused as they present

**Funding:** The author(s) received no specific funding for this work.

**Competing interests:** The authors have declared that no competing interests exist.

primarily as a non-specific febrile illness. Patients with a weakened immune system (e.g. organ transplant patients) are more susceptible to HME and show more complications when compared to patients with a normal immune system. Although response to antimicrobial treatment is usually fast and effective, complications may arise (particularly in patients with a weakened immune system) and the outcome even be fatal. Since most available data on HME comes from case reports, case series and retrospective studies, data on several aspects of the disease remains patchy. To obtain a better overview on various aspect of HME we systematically reviewed the existing literature and compiled and analyzed the reported data on epidemiology, clinical presentation, complications, diagnosis, treatment, and outcome of HME.

## Introduction

Ehrlichioses and anaplasmoses are tick-borne zoonotic infections caused by gram-negative, obligate intracellular bacteria of the family Anaplasmataceae in the order Rickettsiales. All members of the Anaplasmataceae have in common that they survive in vacuoles of host cells, which usually originate from bone marrow, but occasionally also from endothelial cells. The different Anaplasmataceae vary with regard to their tropism for certain cell lines. E.g. *Ehrlichia chaffeensis*, *Ehrlichia canis* and *Ehrlichia muris* infect mostly the monocytes and macrophages, while *Anaplasma phagocytophilum* and *Ehrlichia ewingii* infect mostly the granulocytes of their mammalian hosts. This tropism for certain cell lines is reflected in some of the historical names of the illnesses these pathogens cause in their natural animal as well as accidental human hosts. In the context of human infection this is *human monocytotropic ehrlichiosis* (HME) and *human granulocytotropic anaplasmosis* (HGA).

### Human ehrlichiosis

The first recognition of *Ehrlichia* as a pathogen dates back to 1935, when canine ehrlichiosis caused by *Ehrlichia canis* and its transmission by *Rhipicephalus sanguineus* ticks was discovered. Later studies revealed that two other economically important veterinary pathogens, *Anaplasma marginale* (first described in 1910) and *Cowdria ruminantium* (first described in 1925), are also *ehrlichiae* [1]. The first human infections caused by *Ehrlichia* were reported as early as 1954, when *Rickettsia sennetsu* (later *Ehrlichia sennetsu*, today *Neorickettsia sennetsu*) was identified as the causative agent of Sennetsu fever (a glandular fever-like disease) in Japan [2,3]. Until then, bacteria of the family Anaplasmataceae (today including the genera *Ehrlichia*, *Anaplasma* and *Neorickettsia*) were primarily known as veterinary pathogens. This changed following the identification of *Ehrlichia chaffeensis* (then thought to be *E. canis*) in a blood sample of a 51-year-old man presenting with a febrile illness clinically similar to Rocky Mountain spotted fever after a tick bite in rural Arkansas in 1986 [4,5]. Since then, human monocytotropic ehrlichiosis (HME) due to *E.chaffeensis* has increasingly been recognized as an important tick-borne pathogen in the United States.

In 1999, *E. ewingii* (first described in 1971 [6]), a *Ehrlichia* strain closely related to *E. canis* and *E. chaffeensis* and, like *E. canis* a causative agent of canine ehrlichiosis, was found to also cause illness in humans. These *E. ewingii* infections presented with a clinical picture similar to that of HME caused by *E. chaffeensis* [7]. The natural enzootic cycles of both *E. chaffeensis* and *E. ewingii* primarily involve the Lone Star tick (*Amblyomma americanum*) and the white-tailed deer (*Odocoileus virginianus*) as vector and reservoir. Human infections represent zoonotic spillover events mostly related to recreational, occupational, military-related or peridomestic

**Table 1. The currently known human pathogenic *Ehrlichia* species and their known mammalial hosts and tick vectors.**

| *Ehrlichia* spp. | Mammalian hosts | Major target cell / cell tropism | Reported tick vectors |
|---|---|---|---|
| *E. chaffeensis* | Deer, coyotes, dogs, humans | Monocytes, macrophages | *Amblyomma americanum, Dermacentor variabilis, Ixodes pacificus* |
| *E. canis* | Dogs, humans | Macrophages | *Rhipicephalus sanguineus* |
| *E. ewingii* | Deer, dogs, humans | Granulocytes | *Amblyomma americanum, Dermacentor variabilis* |
| *E. muris* ssp. *muris* and ssp. *eauclairensis* | Mice, voles, (humans*) | Macrophages, endothelial cells? | *Ixodes scapularis, Ixodes persulcatus, Haemaphysalis flava* |
| *E. ruminantium* | Cattle, sheep, goat, wild ruminants, (humans*) | Endothelial cells | *Amblyomma variegatum Amblyomma hebraeum* |
| *Candidatus* E. erythraense | ?, (humans*) | ? | *Haemaphysalis longicornis* |

* to date, only very few isolated human infections have been reported.

exposure to the tick vector [8]. In 1999, HME became a reportable disease in the United States. Between 1987 and 2017, 15,527 cases of HME were reported in the USA [9]. From 2008 to 2012, the incidence of HME was 3.2 cases per million population in the United States, a four-fold increase from 2000 [10]. In areas of high endemicity, incidence rates as high as 138–330 per 100,000 population [11,12] and seroprevalence rates of up to 12.5% are found [11]. In 2009, the spectrum of *Ehrlichia* species causing human ehrlichiosis (HE) in the United States extended to *E. muris subsp. eauclairensis* (formerly *Ehrlichia muris-like* agent), whose enzootic cycle primarily involves the blacklegged deer tick (*Ixodes scapularis*) and the white-footed mouse (*Peromyscus leucopus*) and other small mammals as vector and reservoir [13]. Table 1 shows the currently known human pathogenic *Ehrlichia* species.

Besides infections transmitted by ticks, cases of HME acquired by blood-transfusion [14–16] and organ transplantation have been reported [17,18].

Human ehrlichial infections/pathogens have also been reported in South America, Africa, and Asia, but the epidemiology of HE outside the United States is still largely unknown [19–23].

The taxonomic order of the Anaplasmataceae underwent significant changes over time, with the most relevant one being the identification of *Anaplasma* as an own genus and its separation from the genus *Ehrlichia* in 2001 [1]. With this change of taxonomic order, the previously called human granulocytotropic ehrlichiosis (HGE, first described as a human pathogen presenting similarly to HME in 1994 in the United States [24] became human granulocytotropic anaplasmosis (HGA)). Although human infection caused by all members of the family Anaplasmataceae have been, and often continue to be, generically referred to as ehrlichiosis, it is increasingly apparent that the clinical manifestations and causative agents are distinct [8].

## Clinical presentation

Human ehrlichiosis and human anaplasmosis both present as febrile illnesses manifesting one to two weeks after the bite of an infected tick. The fever is accompanied by unspecific symptoms like weakness, malaise, headache, myalgia, arthralgia, nausea, and vomiting and sometimes a rash is present. Most infections are uncomplicated and self-limiting (and most likely even asymptomatic), but severe complications, including septic shock, acute respiratory failure, meningoencephalitis, renal failure, and multi-organ failure may occur and even be fatal. The higher incidence and risk for life-threatening complications in older patients suggests that host factors are important parameters influencing disease severity. Increased severity and mortality is also observed in immunosuppressed or immunocompromised patients, particularly in HME.

## Diagnostics

Diagnosis of HME and HGA rests primarily on clinical suspicion, epidemiological plausibility, and suggestive unspecific routine blood laboratory results, as the availability of sensitive and specific acute diagnostic tests (i.e. PCR) is often limited. Since the risk for complications increases and prognosis worsens if treatment is delayed, it is recommended to start empirical antimicrobial treatment before the results of the laboratory tests are available [8]. Routine blood laboratory investigations often show cytopenia, particularly leukopenia and thrombocytopenia, elevated liver function test levels (transaminases, alkaline phosphatase), and elevated C-reactive protein levels [25].

Light microscopic examination of blood smears (stained with e.g. Wright's, Diff-Quik, Giemsa) may be diagnostic if the typical, intracellularly located inclusions called morulae (from the Latin word for mulberry) are detected. These morulae represent clustered microcolonies of the pathogen within the host cell vacuole (Fig 1). Although this method is rapid, it is relatively insensitive compared to other confirmatory tests, especially beyond the acute phase (first week of infection) and in immunocompetent patients where disease is usually associated with a very low bacterial burden [8,25]. The sensitivity of microscopy also largely depends on the causative species: morulae are detectable in 25–75% of the blood smear examinations in HGA but only in 3–7% of the blood smear examinations in HME [8,25]. Except for microscopy, PCR assays, serology, immunostaining of biopsy/autopsy material, and cell culture are used to diagnose HME and HGA. Table 2 gives an overview on the advantages, disadvantages and applications of the different diagnostic methods available.

## Treatment and outcome

The treatment of choice for human ehrlichiosis and anaplasmosis is doxycycline or tetracycline and in most cases, clinical response is rapid. The observation that mildly affected patients recover spontaneously even without specific therapy, as well as the high seroprevalence rates reported in some populations, suggest that most infections are mild, self-limiting and to a large extend probably subclinical [8,25].

National surveillance data for HME from the United States show a hospitalisation rate of 57% and an overall death rate of 1%. Higher case fatality rates of 4% in children <5 years and 3% in the elderly ≥70 years are reported [10].

With our systematic review of human ehrlichiosis presented here, we aim to provide clinicians with a comprehensive summary of the available data, focusing on the clinically relevant

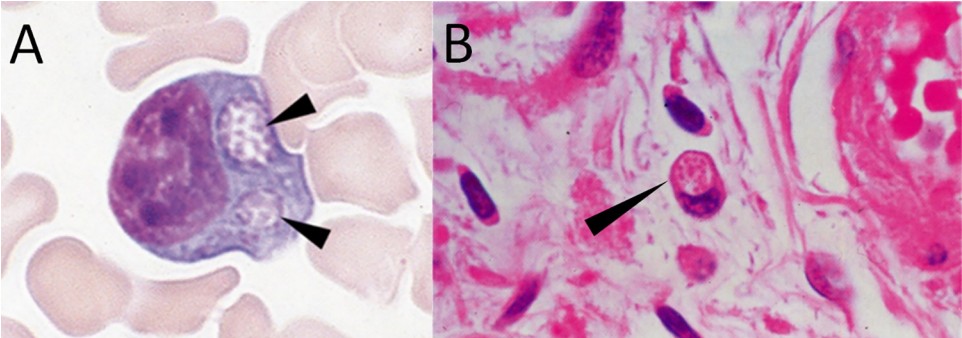

**Fig 1. Microscopical detection of *Ehrlichia chaffensis* morulae.** Intracellular *Ehrlichia chaffeensis* morulae in: (A) light microscopy of the peripheral blood smear preparation (magnification x 1500); (B) leukocyte in the stroma of the ovary (magnification x 1200) [courtesy of Prof. Aileen M. Marty].

**Table 2. Overview of laboratory methods available to diagnose HME and HGA and their advantages, disadvantages and application/use** [26,27].

| Diagnostic method | Advantages | Disadvantages | Application / use |
|---|---|---|---|
| Microscopy of blood smear or buffy coat preparation | Widely available | Limited sensitivity (demands expertise, depends on density of *morulae*); highest sensitivity during acute phase/first week of infection; limited specificity (does not allow conclusive species differentiation) | Used in the acute phase of infection when PCR is not available |
| PCR | High specificity (allows species differentiation); high sensitivity in the acute phase of infection; also suitable for biopsy/ autopsy samples | Decreased sensitivity beyond the acute phase/first week of infection and after administration of appropriate antibiotics | Used for diagnosis in the acute phase of infection; availability often limited to larger/reference laboratories |
| Serology (IFA, ELISA) | Enables retrospective diagnosis beyond the acute phase of infection | Not useful in acute phase of infection (due to delayed seroconversion); confirmation of diagnosis demands paired samples (acute and convalescent serum); limited specificity due to persisting antibodies after infection and cross-reactivity of assays with other rickettsial pathogens; decreased sensitivity after early administration of appropriate antibiotics | Paired serology by IFA is the serological gold standard but the result will only be available retrospectively; used for epidemiological studies |
| Immunostaining of biopsy / autopsy tissue | High specificity; can also be applied to biopsy/ autopsy samples | Availability limited to larger/reference laboratories; decreased sensitivity after administration of appropriate antibiotics | Useful for confirming the diagnosis in fatal cases where diagnostic levels of antibodies did not develop before death |
| Cell culture | High specificity | Low sensitivity; time and resource demanding; decreased sensitivity after administration of appropriate antibiotics | Diagnostic reference standard, but availability largely restricted to reference and research laboratories |

IFA, immunofluorescence assay; ELISA, enzyme-linked immunosorbent assay, PCR, polymerase chain reaction.

core aspects (Note that an analogously compiled systematic review of human granulocytotropic anaplasmosis (HGA) was conducted in parallel by our group).

## Methods

We performed a systematic literature search of the databases EMBASE, CINAHL, Cochrane, PubMed, Scopus and Web of Science on 04/Jan/2023, using the search term (("Ehrlichia" OR "Ehrlichiosis" AND "Monocytes")) OR ("Ehrlichi*" OR "Neoehrlichi*" OR "HME Agent" OR "chaffeensis" OR "ruminantium" OR "muris eauclairensis" OR "muris" OR "sennetsu" OR "ewingii" OR "Ehrlichia canis") NOT ("Animals" NOT "Humans")), adapted to the search format of the different databases. A detailed description of the literature search is available in S1 Text. After removing duplicated by EndNote (Version X9.3.3) and manually, the publications were pre-screened by title and abstract, removing those not concerning HME or not including the objectives of this study (data of clinical cases, including epidemiology, mode of transmission, diagnostic, treatment, outcome). A full-text review of the remaining publications was then performed excluding those not meeting the inclusion criteria, according to the systematic review protocol (concerning HME and the objectives of the study, published in English, German, French, Italian or Spanish), as prescribed in S2 Text to exclude those not meeting the inclusion criteria. Publications that could neither be retrieved through the respective journals, nor by contacting libraries or the corresponding authors, were classified as 'not retrievable' and excluded. During the full-text review, the reference lists of the publications were screened for additional relevant articles not identified previously («snowball-search» strategy). We conducted a second literature search with the identical search strategy on 26/June/2023 to include the most recently published articles. From the finally screened eligible studies, we extracted the following data: author, title, year of publication, type of study, country of study, study period, number of HME cases reported, age, sex, most likely country/province of acquisition/

infection, autochthonous or imported case, if imported: time between end of trip and symptoms, country/province of diagnosis, risk factors for tick bite, year of acquisition/infection, pre-existing conditions, immunosuppression, pregnancy, symptomatic/asymptomatic infection, hospital admission, time between first symptoms and presentation to hospital/physician, duration of hospital stay, signs and symptoms, duration between fever and appropriate treatment, presumed vector of disease, history of tick bite/exposure, tick species, duration between bite and symptoms, diagnostic methods (serology, PCR, microscopy, culture, biopsy), grade of diagnostic certainty, *Ehrlichia* species, coinfections, laboratory data, used drug(s) and treatment regimen(s), number of treated or untreated patients, complications and outcome. Discrepancies and unclear cases were discussed and resolved by consulting a second reviewer. The probability of diagnostic certainty of the individual HME cases was graded according to the diagnostic method used in the included studies, with PCR, culture and immunostaining of tissue having the highest (grade A+) and clinical diagnosis the lowest (grade D) evidence for a correct diagnosis (Table 3). The grading system was derived from the CDC case definition of HME [23]. For interpretation of the laboratory parameters, we used our in-house laboratory reference ranges. For parameters not commonly done in our laboratory, we used the reference ranges of the University Hospital of Basel. The reference table used to interpret the laboratory findings can be found in S3 Text. To limit confounding, patients with coinfections and preexisting conditions were excluded from some of the analyses if it was not clear whether the symptoms, the laboratory abnormalities, complications or the outcome were attributable to HGA.

For analysis, we divided the reviewed clinical cases into two categories, cases reported with individual data (CRID) and cases reported with non-individual data (CRNID), i.e. case series with cumulatively reported/pooled data, and differentiated between HME monoinfections and cases of HME with coinfection(s). The data extraction sheet is available in S4 Text. Data were analyzed and descriptively summarizing in percentages, medians, and ranges. We used the free online geographic application Mapchart (www.mapchart.net) to visualize the worldwide and regional distribution of HME cases and the causative *Ehrlichia* spp. This review was registered on PROSPERO under the number CRD42022385413.

## Results

Our search identified 12,646 publications, of which 264 proved eligible for inclusion in the review (Fig 2). The reference lists of the included and excluded publications, considered publications, and the PRISMA checklist for systematic reviews are available in S5 Text and S6 Text, respectively.

246 of the 264 analyzed publications reported information on 416 CRID. The other 18 analyzed studies reported on cohorts of HME patients with a total of 844 CRNID. For the analysis of signs and symptoms, laboratory findings, complications and outcome, we used data only from cases clearly indicating the causative species as *E. canis* or *E. chaffeensis* monoinfections, thus resulting in 321 CRID and 659 CRNID. The analysis of the data on epidemiology, diagnostics and antimicrobial treatment included mono- and coinfections. Although the analysis of the cases reported with non-individual data (CRNID, i.e. case series, cohorts) was limited, we nevertheless summarized the data analogously to CRID to allow a rough overview and comparison (S7 Text). We also summarized the data on CRID with coinfection(s) (S8 Text), but due to the data heterogeneity and the overall low numbers, we omitted the attempt of specific subgroup analyses. Fig 3 shows the allocation of cases to the respective analysis group.

It should be noted that throughout the further manuscript we use the term HME when specifically referring to human infections with *E. canis* and *E. chaffeensis*, while we use the term HE when referring to human infections with Ehrlichia, regardless of species.

**Table 3. Diagnostic grading system to judge the certainty of the correct diagnosis of HME [26,28,29].**

| Diagnostic method | Description | Grade of diagnostic certainty | Case classification (provided illness clinically compatible with ehrlichiosis) | Comment |
|---|---|---|---|---|
| PCR | Detection of *Ehrlichia* spp. DNA in a clinical specimen via amplification of a specific target by polymerase chain reaction (PCR) assay | A+ | Direct evidence, confirmed diagnosis | High level of evidence, especially in the first week of illness and before start of antibiotics, mostly done from whole blood specimens, also possible in solid tissue and bone marrow specimens |
| Culture | Isolation of *Ehrlichia* spp. from a clinical specimen in cell culture | A+ | Direct evidence, confirmed diagnosis | High level of evidence, especially in the first week of illness and before start of antibiotics, difficult to carry out, time demanding |
| Immunostaining of biopsy/autopsy tissue | Demonstration of ehrlichial antigen in a biopsy/autopsy sample by immunohistochemical methods | A+ | Direct evidence, confirmed diagnosis | High level of evidence, difficult to carry out, time demanding |
| Serology—IgG IFA, paired samples | Serological evidence of a fourfold rise in IgG-specific antibody titer to *E. chaffeensis/E. canis* antigens by indirect immunofluorescence assay (IFA) in paired serum samples (i.e. an acute phase sample [first week of infection] and a convalescent phase sample [2–4 weeks later]) | A | Indirect evidence, confirmed diagnosis | High level of evidence, serological gold standard, cross-reaction with other rickettsial diseases possible |
| Blood smear or buffy coat preparation microscopy | Identification of intracellular *morulae* in monocytes (*E. ewingii*: granulocytes) by microscopic examination | B+ | Direct evidence, probable diagnosis | Intermediate level of evidence, easy to carry out, examiner-dependent, likelihood of detection depends on level of *Ehrlichia* in blood; limited specificity as morulae of *E. ewingii* cannot by differentiated from morulae of *A. phagozytophilum* which also show a tropism for granulocytes |
| Serology—IgG IFA single sample or ELISA | Serological evidence of elevated IgG antibody reactive with *E. chaffeensis/E. canis* antigen by ELISA or IFA (CDC uses an IFA IgG cutoff of ≥1:64) | B | Indirect evidence, probable diagnosis | Intermediate level of evidence, no certain differentiation between acute and past infection possible, cross-reaction with other rickettsial diseases possible |
| Serology—IgM ELISA or IFA | Serological evidence of elevated IgM antibody reactive with *E. chaffeensis/E. canis* antigen by IFA, ELISA, or assays in other formats | C | Indirect evidence, possible diagnosis | Low level of evidence, IgM tests are not always specific and not useful as single means of diagnosis |
| Clinical diagnosis | Signs and symptoms compatible with ehrlichiosis | D | Clinical diagnosis | The lowest level of evidence for correct diagnosis |

CDC, Centers for Disease Control and Prevention; DNA, deoxyribonucleic acid; ELISA, enzyme-linked immunosorbent assay; IFA, immunofluorescence assay; IgG, immunoglobulin G; IgM, immunoglobulin M; PCR, polymerase chain reaction.

## Epidemiology

Fig 4 and Table 4 show the number of publications reporting cases of human ehrlichiosis, the number of CRID and CRNID, and the geographic origin of these reports from 1987 to 2023.

Table 5 lists the number of CRID and CRNID reported from the different regions of the world. Fig 5 shows the worldwide distribution of reported HE cases. Fig 6 shows the distribution of HE cases in the USA according to national surveillance data and according to the data analyzed for this review.

Fig 7 shows the number of HE due to *E. canis* and *E. chaffeensis*. The red line marks the year in which it was discovered that in humans the culprit *Ehrlichia* species is *E. chaffeensis* and not *E. canis* [4].

Of the 264 analyzed publications, 12 publications reported a total number of 61 non-*canis*/non-*chaffeensis* HE cases (Table 6). In 32 cases of HE, a co-infection was present. The most common co-infection was with Rickettsia spp., the 2nd most common with *Borrelia burgdorferi*

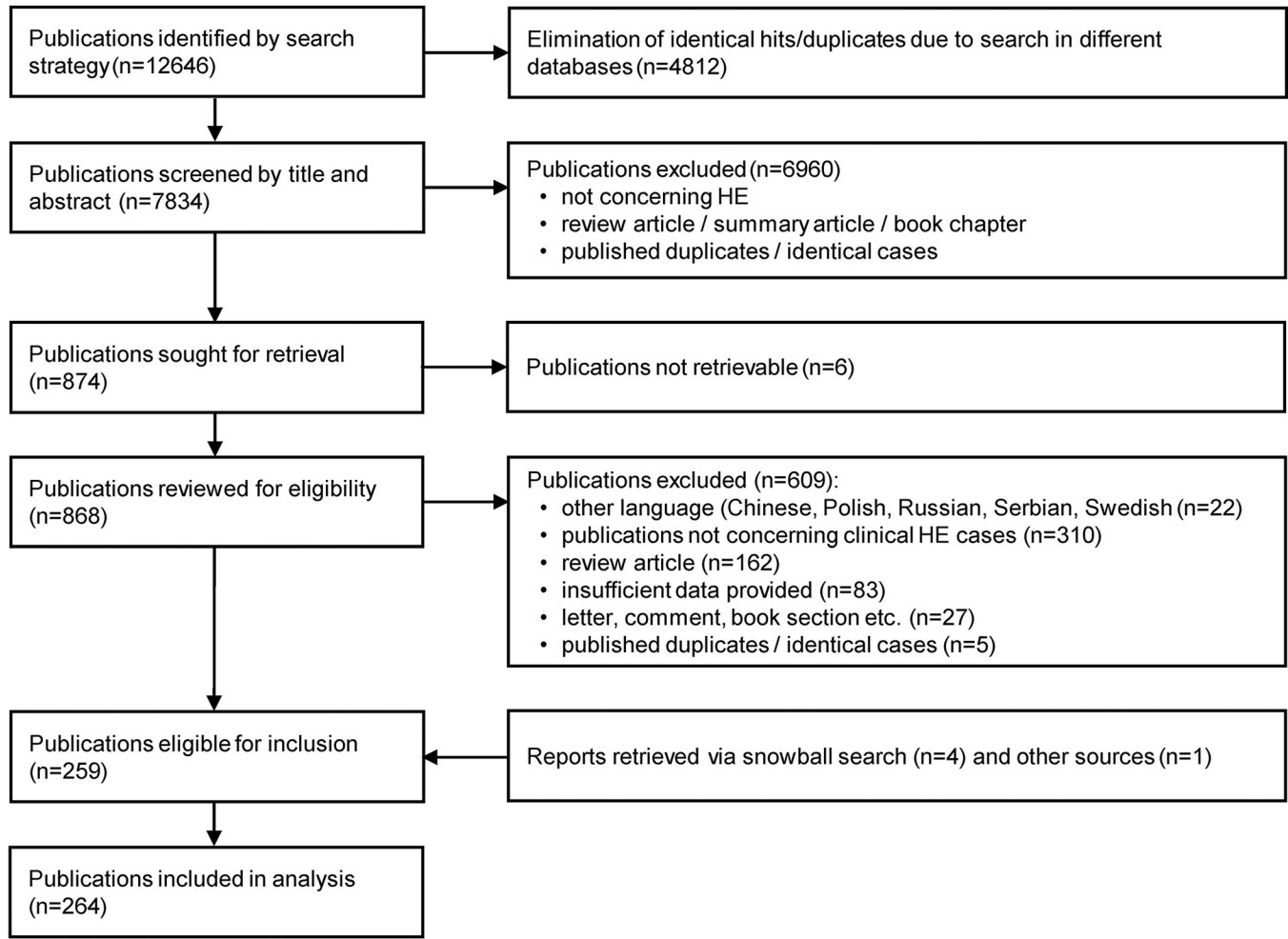

**Fig 2. Flow diagram of search and selection of eligible publications.** HE, human ehrlichiosis.

(Table 7). Data on the diagnostic methods used to diagnose HE was available for 413 CRID and 814 CRNID, the majority of cases having been diagnosed with PCR (Table 8).

In 66 cases, the detection of morulae by light microscopy was described. In 43 of these cases (65.2%), the diagnosis of HE was confirmed by PCR and in 32 of the cases the cell line containing the morulae was specified (Table 9).

The use of cell culture was reported in 17 cases. In 15 cases, a DH82 cell line was used [30,41] and in two cases the used cell line was not specified [42,43].

### Analysis of HME infection cases reported with individual data (CRID)

For 414 cases, the patient's sex was reported: 268 (64.7%) were male, 146 (35.3%) were female. For 409 cases (264 male, 145 female) sex and age was reported: the median age (range) of male and female patients was 45.5 (2–85) and 50 years (2–95), respectively.

For 111 cases (26.7%), preexisting immunocompromization/-suppression was reported: 92 were on immunosuppressive medication, 61 of them due to a solid organ transplantation; 14 were HIV positive, 10 of them with a reported low $CD_4$ cell count or AIDS defining conditions; three had a splenectomy; one chronic renal failure; one was described as immunosuppressed by the authors without further explanation.

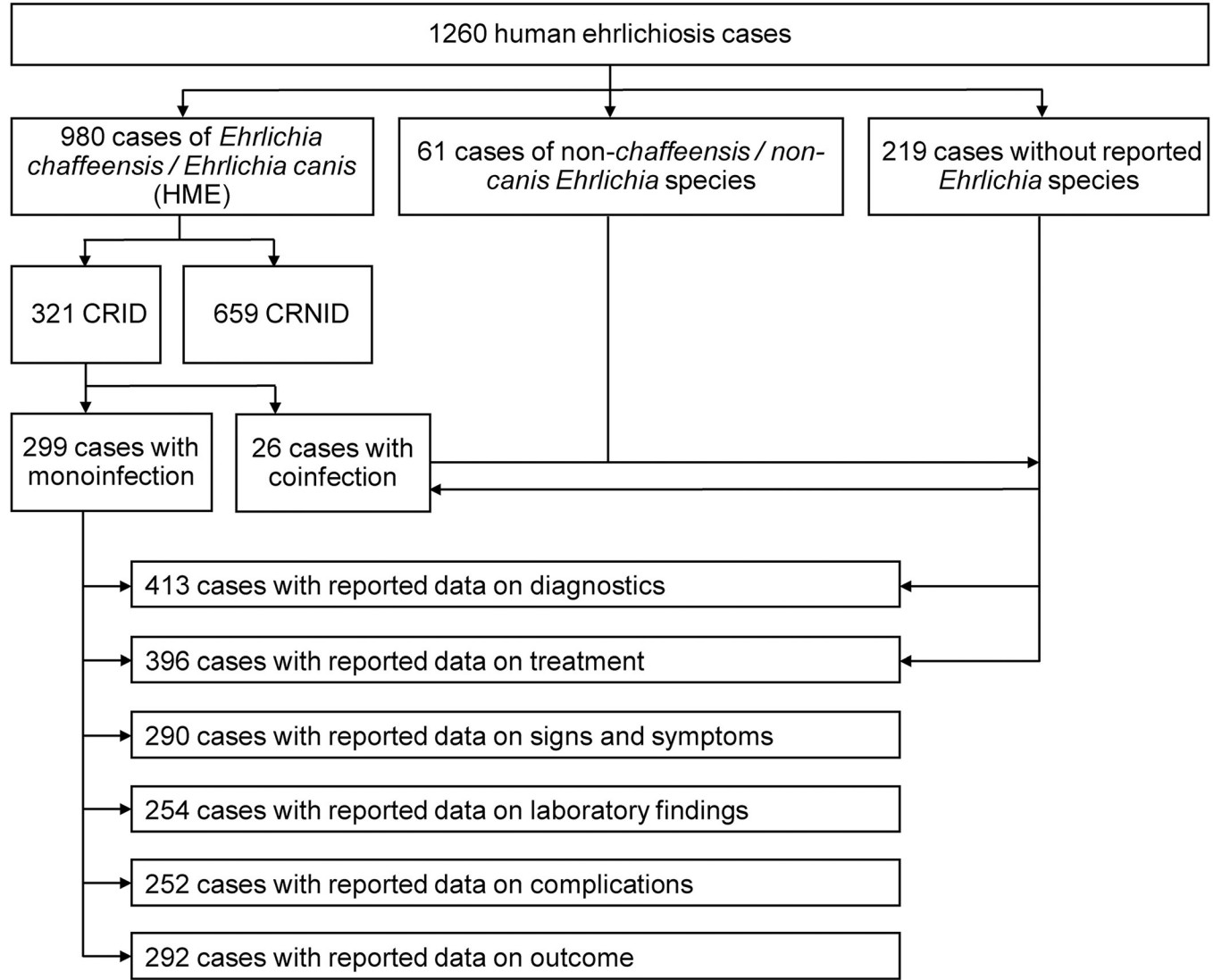

**Fig 3. Allocation of the reviewed human ehrlichiosis cases to the respective analysis groups.** CRID, cases reported with individual data; CRNID, cases reported without individual data; HME, human monocytotropic ehrlichiosis.

The suspected route of transmission was tick-borne in 245 (96.1%) cases. Among 210 patients actively assessed for the history of a tick bite, 152(72.4%) recalled a tick bite. The culprit tick species was only mentioned in two cases, in both cases it was *Amblyomma americanum* (the Lone star tick). In 114 (97.4%) of 117 cases with respectively available data, outdoor activities were reported as risk factor for HME. The reported outdoor activities were recreational (camping, hiking, biking, fishing, playing golf, gardening) in 73% and occupational (most commonly work in construction, on farms or military training, rarely field biology) in 27% of the cases. Other reported transmission routes included blood transfusion (n = 3; Table 10) and organ transplantation (n = 7; Table 11).

Two cases of HE related to international travel were reported (Table 12).

One case of HE during pregnancy was reported. The woman was 13 weeks pregnant. She recovered completely and both mother and infant showed no long-term sequelae during the 1 year of follow-up. No data on whether vertical transmission occurred was reported [46].

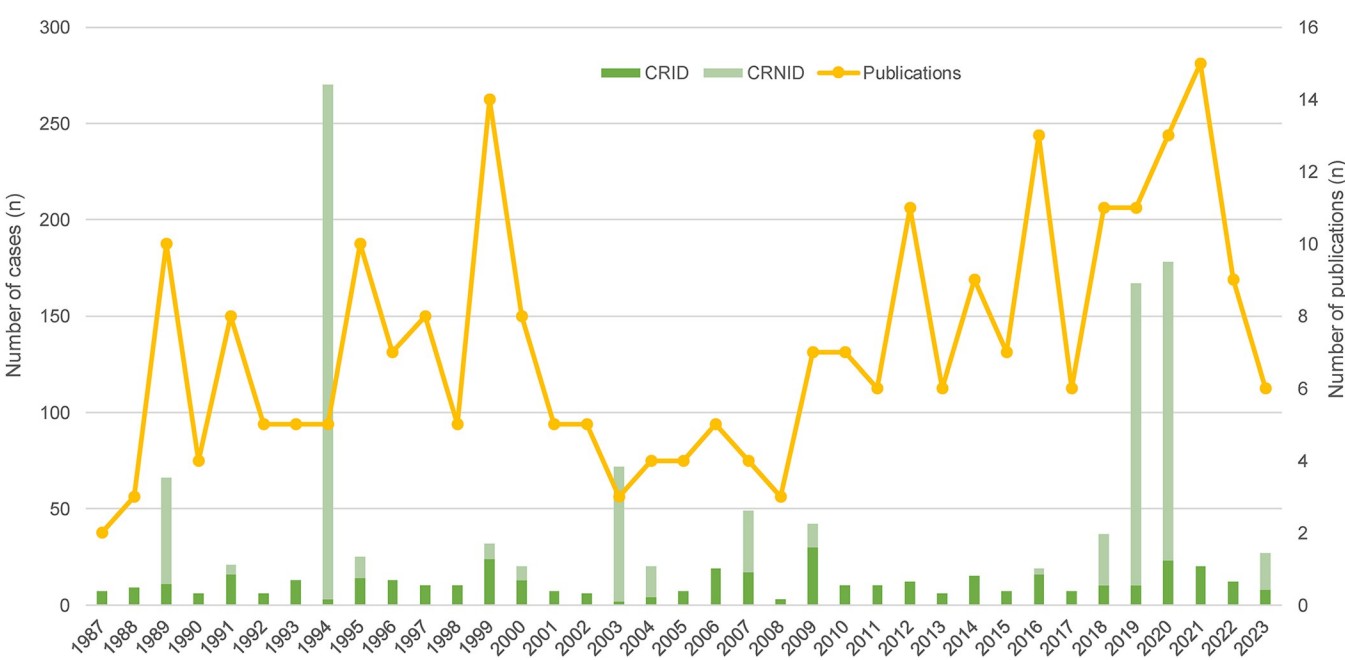

**Fig 4. Number of publications reporting cases of human ehrlichiosis and the respective number of cases published 1987–2023.** CRID, cases reported with individual data; CRNID, cases reported with non-individual data.

For 299 HME monoinfection CRID the information whether the patients were symptomatic or asymptomatic was reported: 298 (99.7%) were symptomatic, 1 (0.3%) was asymptomatic. For 175 (58.7%) HME monoinfection CRID, data on the timespan from symptom onset to presenting to a physician or hospital was available: median 5 days (range 0–28 days). Data on the incubation period in case of tick-borne transmission was available for 72 cases: median 14 days (range 1–66 days). Data on the incubation period in case of transmission by organ transplantation was available for all 7 reported cases: median 11 days (range 9–25 days). Data on the incubation period in case of transmission by blood product transfusion was only available for only one of the 3 reported cases: 22 days (Table 10).

Hospitalization was reported for 248 (83.2%) of the symptomatic cases. Data on the length of hospitalization was available for 76 patients: median eight 8 days (range 1–127).

## Signs and symptoms

For 290 of the 299 HME monoinfection CRID, data on signs and symptoms was available (Fig 3). Fig 8 shows the frequency of the most commonly reported symptoms. Information on the exact duration of fever was available for 39 patients: median 8 days (range 1–120), with five patients having a persistent or recurring fever for more than 20 days. For 153 cases, data on the maximum temperature measured was available: median 39.4˚C (range 36.3–41.6˚C).

The presence of a rash was reported in 71 (24.5%) cases. In children (i.e. <18 years), 27 out of 62 case patients (43%) presented with a rash. In adults, 44 out of 226 case patients (19%) presented with a rash. In 48 (67.6%) of these cases, information on the rash morphology was available: macular (29.1%), petechial (25%), erythematous (20.8%), maculopapular (14.6%). Less frequently, the rash was described as: purpuric (4.2%), vasculitic (2.1%), nodular (2.1%) or lacy (2.1%).

**Table 4. Number of publications reporting human ehrlichiosis cases and number of reported cases by country 1987–2023.**

| Country | Number of publications reporting human ehrlichiosis cases n (%) | Number of reported human ehrlichiosis cases n (%) |
|---|---|---|
| USA | 236 (89) | 1173 (93) |
| China | 2 (<1) | 20 (2) |
| Cameroon | 1 (<1) | 12 (1) |
| Venezuela | 2 (<1) | 7 (1) |
| Israel | 2 (<1) | 7 (1) |
| Belgium | 1 (<1) | 1 (<1) |
| Brazil | 2 (<1) | 11 (<1) |
| Canada | 1 (<1) | 1 (<1) |
| Colombia | 3 (1) | 3 (<1) |
| Japan | 1 (<1) | 3 (<1) |
| Mali | 0 (0) | 1 (<1) |
| Mexico | 2 (<1) | 3 (<1) |
| Nicaragua | 1 (<1) | 1 (<1) |
| Portugal | 1 (<1) | 1 (<1) |
| Serbia | 1 (<1) | 1 (<1) |
| South Africa | 1 (<1) | 3 (<1) |
| Spain | 1 (<1) | 1 (<1) |
| Taiwan | 2 (<1) | 2 (<1) |
| Thailand | 1 (<1) | 3 (<1) |
| Tunisia | 1 (<1) | 5 (<1) |
| Türkiye | 1 (<1) | 1 (<1) |
| Australia | 1 (<1) | 0 (0) |
| Total | 264 (100) | 1260 (100) |

USA, United States of America.

## Laboratory findings

Of the 299 HME monoinfection CRID, data on laboratory findings was available for 254 cases (Fig 3). Fig 9 shows the frequency and Table 13 the median values and ranges of the reported abnormal laboratory findings.

## Complications

Complications were reported in 159 (63.1%) of 252 HME monoinfection CRID (Fig 3). Fig 10 shows the most frequently reported complications among the 159 cases. Exacerbation of an

**Table 5. Number of human ehrlichiosis cases by geographic region.**

| Geographic region | Number of CRID n (%) | Number of CRNID n (%) |
|---|---|---|
| North America | 373 (90) | 805 (95) |
| South America | 21 (5) | 0 (0) |
| Asia | 13 (3) | 22 (3) |
| Europe | 5 (1) | 0 (0) |
| Africa | 4 (1) | 17 (2) |
| Total | 416 (100) | 844 (100) |

CRID, cases reported with individual data; CRNID, cases reported with non-individual data.

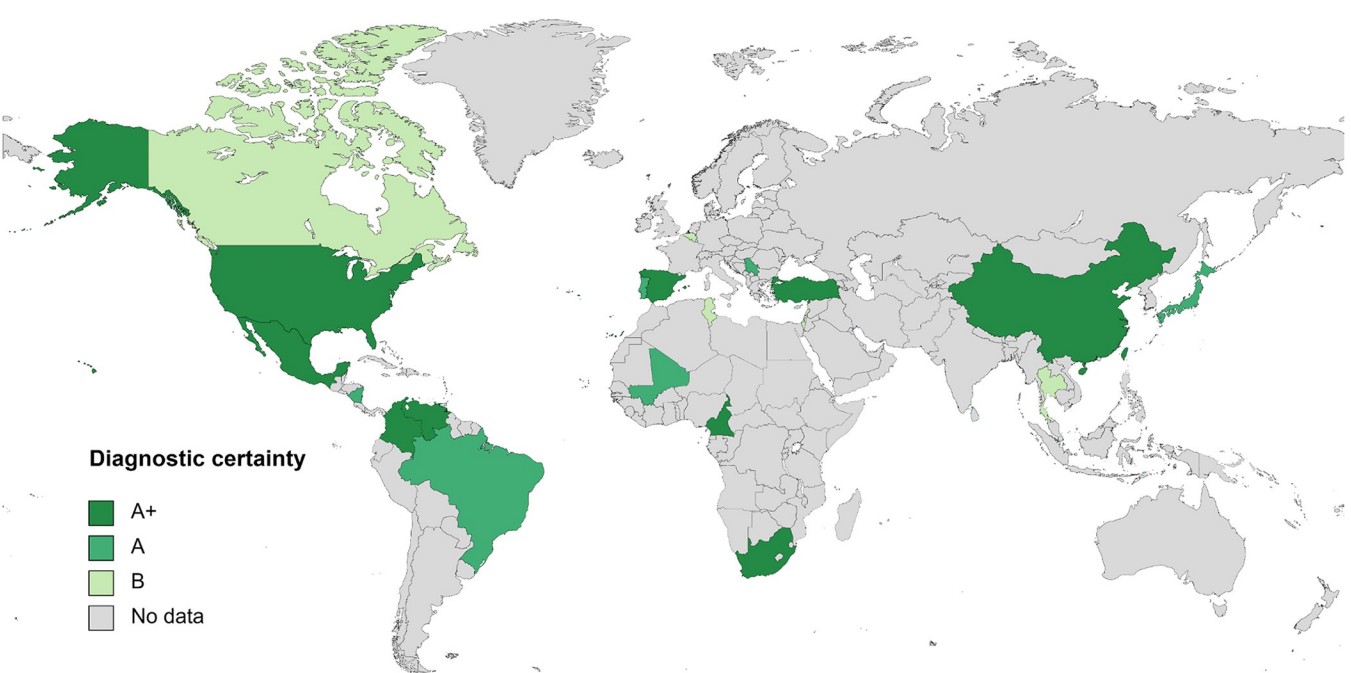

**Fig 5. Reported human ehrlichiosis cases by country.** A+, diagnosed by PCR, culture and/or immunostaining of biopsy/autopsy tissue; A, diagnosed by paired IgG IFA serology; B, diagnosed by single IgG IFA or ELISA serology.

underlying medical condition was not considered a complication of HME (e.g. exacerbation of COPD).

Hemophagocytic lymphohistiocytosis (HLH) was reported in 64 cases: 43 HME monoinfection CRID and 21 non-*canis*/non-*chaffeensis* HE cases and/or in cases with coinfection (Table 14).

## Treatment

Data on antimicrobial treatment was available for 396 of the 416 analyzed HE CRID: 381 cases (96.2%) received antimicrobial treatment, 11 (2.8%) cases received no antimicrobial treatment,

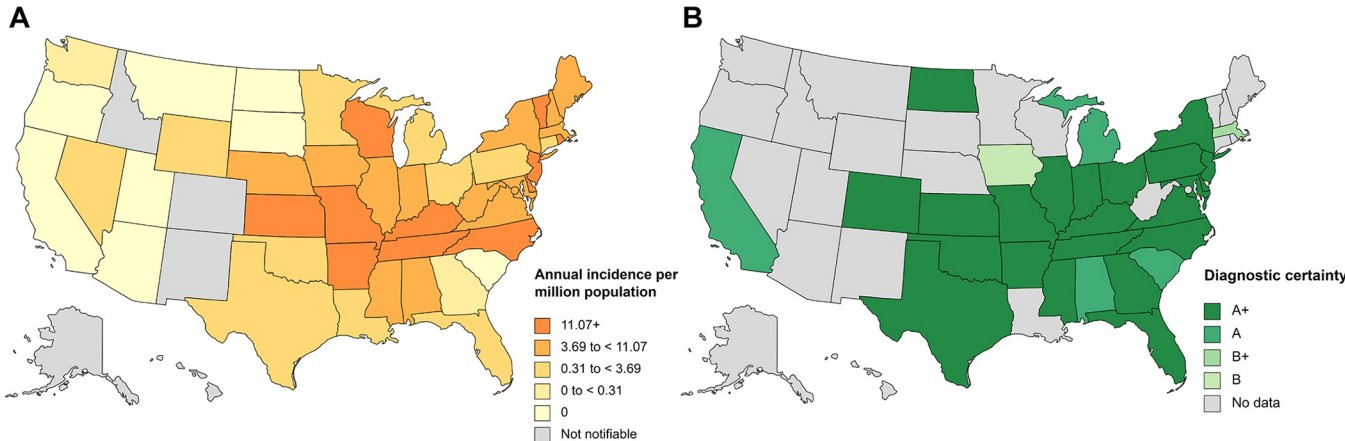

**Fig 6. Distribution of human ehrlichiosis in the United States of America.** (A) Annual incidence (per million population) of reported *Ehrlichia chaffeensis* ehrlichiosis for 2019 [9]; (B) Reviewed cases according to diagnostic certainty; A+, diagnosed by PCR, culture and/or immunostaining of biopsy/autopsy tissue; A, diagnosed by paired IgG IFA serology; B+, diagnosed by microscopy of blood smear or buffy coat preparation; B, diagnosed by single IgG IFA serology.

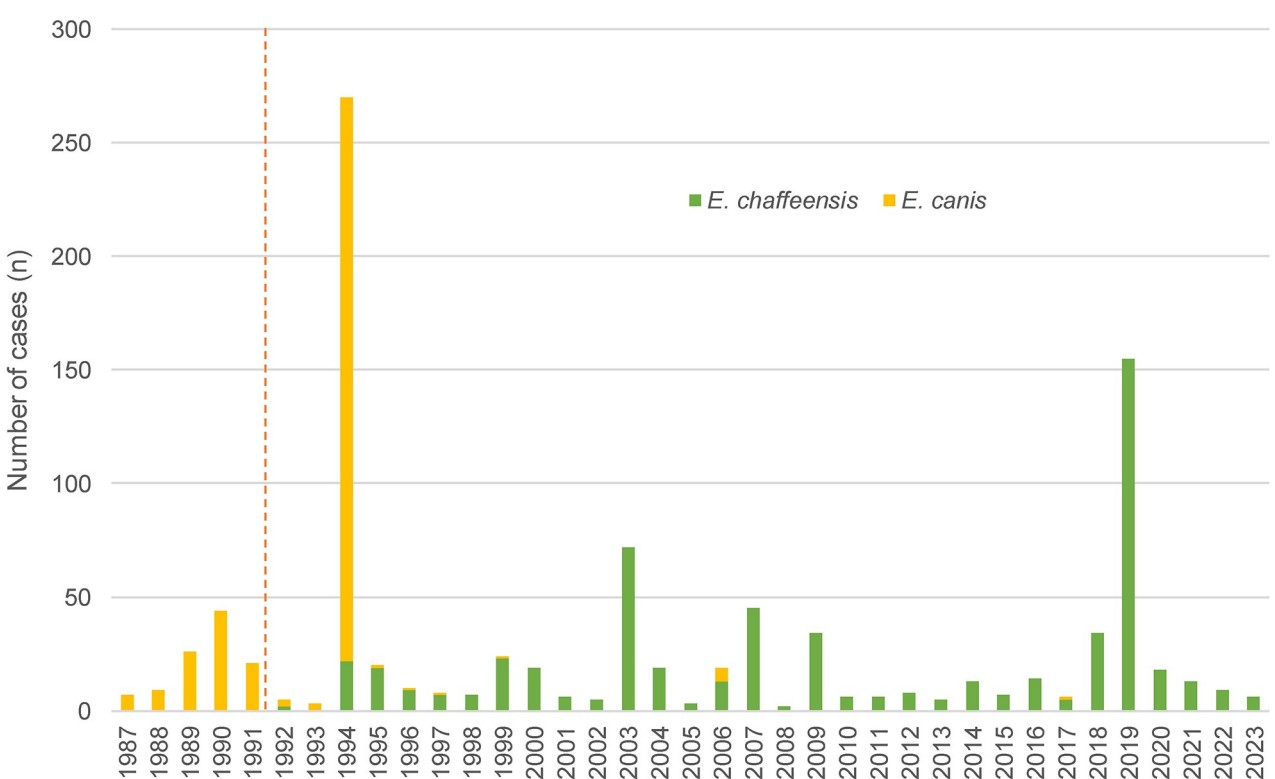

**Fig 7. Diagnosed human ehrlichiosis cases by species before and after the discovery of *Ehrlichia chaffeensis*.** E. chaffeensis, Ehrlichia chaffeensis; E. canis, Ehrlichia canis.

and in four cases (1%) the antimicrobial treatment was not specified. Of the 381 cases receiving antimicrobial treatment, 365 (95.8%) received appropriate antimicrobial treatment (* substances listed in Table 15). One case was treated with tigecycline, which was considered adequate for the treatment of HE by the authors [18].

For 88 of the 365 appropriately treated patients, the time between presentation to a physician or hospital and administration of appropriate therapy was known: median 2 days (range 0–38 days). In 235 of the 365 appropriately treated cases (64.4%), antimicrobial treatment was initiated empirically, in 37 cases (10.1%) antimicrobial treatment was initiated after receiving a positive diagnostic result for HE, and for 93 cases (25.5%), information was not available. Data on the time window between symptom onset and administration of appropriate antimicrobial treatment was available for 62 cases: median 7 days (range 0–59).

Data on the time window between first administration of appropriate antimicrobial treatment and the resolution of fever was available for 72 cases: median time 2 days (range <1–6 days). In one case, treatment was only started after the fever had already subsided. In the majority of cases, doxycycline in the standard dose of 200 mg per day (100mg BID) was given. The overall median duration of antimicrobial treatment was 14 days (range 4–30 days) (Fig 11).

## Outcome

Data on the outcome was available for 292 of the HME CRID: 34 (11.6%) died due to acute complications related to their infection (Table 16), 21 of them were immunocompetent, 13 were immunosuppressed, resulting in case fatality rates of 9.9% and 16.3% respectively. Information on the time between first symptoms and death due to ehrlichiosis was reported in 19 cases: median 13 days (range 7–68 days). Of the survivors with respectively available data, 11

**Table 6. Reported cases of human ehrlichiosis due to non-*canis*/non-*chaffeensis Ehrlichia* species (n = 61).**

| Species | Number of cases (n) | Country of infection | Year of publication | Mode of diagnosis (n patients) | Clinical features (n patients) | Laboratory findings (n patients) | Outcome | Comment | Ref. |
|---|---|---|---|---|---|---|---|---|---|
| *E. ewingii* | 4 | USA | 1999 | PCR (4/4), blood smear (2/4) | Fever (4/4), headache (4/4), myalgia (1/4), neck stiffness (1/4), lymphadenopathy (1/4) | Thrombocytopenia (4/4), anemia (2/4), leukopenia (1/4), abnormal liver-function tests (1/4) | All survived | 3/4 immunocompromised | [7] |
| | 1 | USA | 2004 | PCR, blood smear, culture | N.r. | N.r. | N.r. | | [30] |
| | 3 | USA | 2007 | PCR (3/3) | Fever (3/3), headache (2/3), nausea and vomiting (2/3), myalgia (1/3), cough (1/3) | N.r. | All survived | 3/3 immunocompromised | [31] |
| | 1 | USA | 2009 | PCR | Fever, chills, myalgias, nausea and headache | Leukopenia, elevated liver enzymes | Survived | No cross-reactivity with *E. chaffeensis* | [32] |
| | 7 | USA | 2009 | PCR (7/7) | Fever (7/7), headache (3/7), cough (2/7), malaise (2/7), rash (1/7), arthralgias (1/7), weakness (1/7), nausea (1/7), dyspnea (1/7) | N.r. | All survived | 7/7 immunocompromised | [33] |
| | 1 | USA | 2013 | PCR, blood smear | Fever, fatigue, vomiting, diarrhea, petechial rash | Neutropenia, thrombocytopenia, elevated liver enzymes | Survived | *E. ewingii* infection through platelet transfusion, immunocompromised | [14] |
| | 1 | USA | 2014 | PCR, blood smear | Fever, cough, myalgia, fatigue, weakness, petechiae and bruising | Pancytopenia, elevated bilirubin, LDH and ferritin | Survived | Involvement of bone marrow | [34] |
| | 5 | USA | 2016 | PCR (5/5) | Fever (5/5), headache (2/5), nausea (2/5), malaise (1/5), rash, myalgia(1/5), arthralgia (1/5), vomiting (1/5), abdominal pain (1/5), diarrhea (1/5) | Leukopenia (4/5), thrombocytopenia (4/5), elevated liver enzymes (2/5) | 4/5 survived, 1/5 died of HE unrelated cause | | [35] * |
| | 1 | USA | 2017 | PCR, blood smear | Fever, chills, nausea, vomiting, pedal edema | Pancytopenia | Survived | Immunocompromised | [36] |
| | 10 | USA | 2019 | PCR (10/10) | N.r. | N.r. | 2/10 died, 8/10 survived | 7/10 immunocompromised | [37] |
| *E. ruminantium* | 3 | South Africa | 2005 | PCR (3/3) | Encephalitis (2/3), 1/3 fever, headache, ataxia, progressive sleepiness | Leukocytosis and thrombocytosis (1/3) | All died | | [38] |
| Panola Mountain *Ehrlichia* | 1 | USA | 2008 | PCR | Neck pain | N.r. | Survived | | [39] |
| *Ehrlichia* sp. Wisconsin HM543745 | 4 | USA | 2011 | PCR (4/4), culture (1/4) | Fever (4/4), headache (4/4), malaise/fatigue (4/4), nausea and vomiting (1/4) | Lymphopenia (4/4), thrombocytopenia (3/4), elevated liver enzymes (3/4) | All survived | 2/4 immunocompromised | [13] |

(*Continued*)

**Table 6.** (Continued)

| Species | Number of cases (n) | Country of infection | Year of publication | Mode of diagnosis (n patients) | Clinical features (n patients) | Laboratory findings (n patients) | Outcome | Comment | Ref. |
|---------|---------------------|----------------------|---------------------|--------------------------------|-------------------------------|----------------------------------|---------|---------|------|
| *Candidatus* Ehrlichia erythraense | 19 | China | 2023 | PCR (15/19), paired IgG-IFA serology (4/19) | 19/19 fever, rash, asthenia and anorexia, 15/19 myalgia | 6/19 leukopenia, thrombocytopenia, elevated liver enzymes, 18/19 increased CRP | All survived | | [40] |

CRP, C-reactive protein; *E. ewingii*, *Ehrlichia ewingii*; *E. ruminantium*, *Ehrlichia ruminantium*; HE, human ehrlichiosis; IgG, immunoglobulin G; IFA, immunofluorescence assay; LDH, lactate dehydrogenase; N.r., none/not reported; PCR, polymerase chain reaction; Ref., reference; USA, United States of America.

* regarding [35]: this publication describes an additional five cases of human *Ehrlichia ewingii* infections. Those cases were not included in our analysis because of insufficient clinical data.

(3.8%) were reported to suffer from sequelae, 4.2% in the immunocompetent group and 2.5% in the immunocompromised group (Table 17).

## Discussion

### Publications on HME

Since HME was first described in 1987, the number of publications reporting cases of HME has increased slightly over time, with large cohort studies published in 1994, 2019 and 2020 each providing data on more than 100 cases (Fig 4) [37,115,116].

**Table 7. Reported coinfections in human ehrlichiosis (n = 32).**

| Reported coinfection pathogen(s)* | Number of coinfections among human ehrlichiosis CRID [n = 416] n (%) | Number of coinfections among human ehrlichiosis CRNID [n = 844] n (%) | Number of coinfections among all human ehrlichiosis cases [n = 1260] n (%) |
|-----------------------------------|----------------------------------------------------------------------|------------------------------------------------------------------------|----------------------------------------------------------------------------|
| **Tick-borne pathogens** | | | |
| *Borrelia burgdorferi* sensu lato | 7 (1) | 1 (<1) | 8 (<1) |
| Colorado tick fever | - | 1 (<1) | 1 (<1) |
| *Borrelia burgdorferi* s.l. + *Babesia* spp. | 1 (<1) | - | 1 (<1) |
| *Babesia* spp. | 2 (<1) | - | 2 (<1) |
| *Rickettsia* spp. | 6 (1) | 4 (<1) | 10 (<1) |
| *Rickettsia rickettsii* + *Coxiella burnetti* | 1 (<1) | - | 1 (<1) |
| *Anaplasma phagocytophilum* | 3 (<1) | - | 3 (<1) |
| **Other arthropod-borne pathogens** | | | |
| *Orientia tsutsugamushi* | 1 (<1) | - | 1 (<1) |
| Dengue virus | 1 (<1) | - | 1 (<1) |
| **Non-vector-borne pathogens** | | | |
| *Escherichia coli* | 1 (<1) | - | 1 (<1) |
| Epstein-Bar virus | 3 (<1) | - | 3 (<1) |
| Total | 26 | 6 | 32 |

CRID, cases reported with individual data; CRNID, cases reported with non-individual data; spp. species.

* Note that the list of coinfections is based on the coinfections reported by the authors in the original publication, which, however, often cannot by checked for validity due to a lack of information on the corresponding diagnostics and therefore do not allow a conclusive assessment.

**Table 8. Diagnostic methods used to diagnose human ehrlichiosis.**

| Diagnostic method | Level of diagnostic certainty* | Human ehrlichiosis CRID [n = 413] | | Human ehrlichiosis CRNID [n = 814] | |
|---|---|---|---|---|---|
| | | Number of cases tested positive by this method n (%)# | Number of cases for which this level of diagnostic certainty was the highest n (%) | Number of cases tested positive by this method n (%)# | Number of cases for which this level of diagnostic certainty was the highest n (%) |
| PCR | A+ | 216 (52.3) | 219 (53) | 432 (53.1) | 432 (53.1) |
| Culture | | 3 (<1) | | 14 (1.7) | |
| Immunostaining of biopsy tissue | | 16 (3.9) | | - | |
| Serology–IgG IFA, paired samples | A | 109 (26.4) | 92 (22.3) | 341 (41.9) | 308 (37.8) |
| Microscopy | B+ | 53 (12.8) | 12 (2.9) | 13 (1.6) | 2 (<1) |
| Serology–IgG IFA, single sample | B | 50 (12.1) | 33 (8.0) | 86 (10.6) | 69 (8.5) |
| Serology–IgG ELISA, single sample | | - | | - | |
| Serology–IgM IFA or ELISA | C | 8 (1.9) | 53 (12.8) | - | 3 (<1) |
| Serology–method not specified | | 70 (16.9) | | 28 (3.4) | |
| Clinical diagnosis | D | 4 (<1) | 4 (1) | - | - |

CRID, cases reported with individual data; CRNID, cases reported with non-individual data; ELISA, enzyme-linked immunosorbent assay; IFA, immunofluorescence assay; PCR, polymerase chain reaction.

* A+, diagnosed by PCR, culture and/or immunostaining of biopsy/autopsy tissue; A, diagnosed by paired IgG IFA serology; B+, diagnosed by microscopy of blood smear or buffy coat preparation; B, diagnosed by single IgG IFA or ELISA serology; C, diagnosed by IgM serology and/or unspecified serology method; D, clinically diagnosed.

# in many cases a combination of diagnostic tests was used to establish the diagnosis. Thus, the number of positive test results exceeds the number of cases.

## Epidemiology

The vast majority of HE cases are reported from the USA (Table 4 and 5). Until 1991, it was assumed that cases of HME in the US were exclusively caused by *E. canis*. Then, in 1991, Anderson et al. identified a new *Ehrlichia* species in the US, closely related to but different from *E. canis*, which was named *E. chaffeensis*. Since then, the majority of published HME cases is reported to be due to *E. chaffeensis*, with only very few publications reporting cases due to *E. canis*. This led to the recognition that the main causative pathogen of HME is *E. chaffeensis* and and not *E. canis* [4]. This paradigm change is nicely illustrated in Fig 4.

Very few publications report HE cases from regions of the world other than North America. Most of these cases are based on low diagnostic certainty regarding the correct diagnosis of

**Table 9. Cell line(s) in which *Ehrlichia* morulae were reported (n = 66).**

| Cell line | Number of cases n (%) | *Ehrlichia* species |
|---|---|---|
| Monocytes | 22 (33.3) | 18 *E. chaffeensis*, 1 *E. canis*, 3 unspecified *E.* spp. |
| Granulocytes | 9 (13.6) | 5 *E. ewingii*, 3 *E. chaffeensis*, 1 unspecified *E.* spp. |
| Thrombocytes | 1 (1.5) | *E. chaffeensis* |
| Unspecified and/or multiple cell lines and/or morulae in bone marrow biopsy | 34 (51.5) | |

**Table 10. Reported cases of blood product transfusion-transmitted human ehrlichiosis (n = 3).**

| No. | Year of publication | Age of patient (years) | Sex of patient | Country of infection | Pre-existing medical condition | Immunosuppressive treatment | Culprit blood product | Indication for blood product transfusion | Storage time of blood product (days) | Time between transfusion and onset of symptoms (days) | Diagnosis of human ehrlichiosis established by | Ehrlichia species | Antimicrobial therapy | Time between onset of fever and appropriate treatment (days) | Complications | Outcome | Ref. |
|---|---|---|---|---|---|---|---|---|---|---|---|---|---|---|---|---|---|
| 1 | 2013 | 9 | Male | USA | ALL | Chemo-therapy | Thrombocytes | ALL | 5 | 22 | PCR, microscopy | E. ewingii | Doxycycline | 11 | N.r. | Survived | [14] |
| 2 | 2018 | 59 | Female | USA | AML | Chemo-therapy, radiation | RBC, thrombocytes | AML | N.r. | N.r. | PCR, microscopy | E. chaffeensis | Doxycycline | 15 | Pneumonitis, myocarditis, acute kidney injury, graft loss | Survived | [15] |
| 3 | 2020 | 37 | Male | Mexico | Femoral head fracture | N.r. | N.r. | Bleeding during surgery | N.r. | N.r. | PCR | E. chaffeensis | N.r. | N.r. | Septic shock | Died | [16] |

ALL, acute lymphoblastic leukemia; AML, acute myelogenous leukemia; AKI, acute kidney injury; E. chaffeensis, Ehrlichia chaffeensis; E. ewingii, Ehrlichia ewingii; N.r., none/not reported. PCR, polymerase chain reaction; RBC, red blood cells; Ref, reference; USA, United States of America.

**Table 11. Reported cases of organ transplant-donor derived human ehrlichiosis (n = 7).**

| No. | Year of publication | Age of patient (years) | Sex of patient | Country of infection | Pre-existing medical conditions | Organ transplantation due to | Culprit transplant organ | Time between transplantation and onset of symptoms (days) | Diagnosis of human ehrlichiosis established by | Ehrlichia species | Antimicrobial therapy | Time between onset of fever and appropriate treatment (days) | Complications | Outcome | Ref. |
|---|---|---|---|---|---|---|---|---|---|---|---|---|---|---|---|
| 1 | 2014 | 57 | Female | USA | ESRD secondary to hypertension, HCV, GERD | ESRD | Kidney | 22 | PCR, serology | E. chaffeensis | Doxycycline | 5 | N.r. | N.r. | [18] |
| 2 | 2014 | 56 | Male | USA | HTN, HCV, sleep apnea, | HTN | Kidney | 25 | Serology | E. chaffeensis | Tigecycline | 12 | Renal failure, DIC, spontaneous hemorrhage | Survived | [18] |
| 3 | 2021 | 70 | Male | USA | ESRD from polycystic kidney disease | ESRD | Kidney | 17 | PCR, microscopy | E. chaffeensis | Doxycycline | 7 | HLH, seizure, E. coli septicemia, DVT, candida fungemia | Died | [17] |
| 4 | 2021 | 66 | Male | USA | ESRD due to DM2, nephrectomy for renal cell carcinoma, adult onset Still disease | ESRD, renal cell carcinoma | Kidney | 9 | PCR | E. chaffeensis | N.r. | N.r. | HLH, cardiac arrest | Died | [17] |
| 5 | 2021 | 5 | Male | USA | ESRD due to obstructive uropathy | ESRD | Kidney | 9 | PCR | N.r. | Doxycycline | 7 | N.r. | Survived | [17] |
| 6 | 2021 | 6 | Male | USA | ESRD due to obstructive uropathy | ESRD | Kidney | 9 | PCR | E. chaffeensis | Doxycycline | 9 | HLH | Survived | [17] |
| 7 | 2021 | 69 | Male | USA | HCV-related cirrhosis | HCV-related cirrhosis | Liver | 11 | PCR | N.r | Doxycycline | 1 | N.r. | Survived | [17] |

DIC, disseminated intravascular coagulopathy; DM2, diabetes mellitus type 2; E. coli, Escherichia coli; E. chaffeensis, Ehrlichia chaffeensis; ESRD, end-stage renal disease; DVT, deep vein thrombosis; GERD, gastroesophageal reflux disease; HCV, hepatitis C virus; HLH, hemophagocytic lymphohistiocytosis; HTN, hypertensive nephropathy; N.r., none/not reported; PCR, polymerase chain reaction; Ref., reference; USA, United States of America.

**Table 12. Reported cases of travel-related human ehrlichiosis (n = 2).**

| No. | Year of publication | Number of cases (n) | Country of infection | Country of diagnosis | Level of diagnostic certainty of human ehrlichiosis | Coinfections | Ref. |
|-----|------|------|------|------|------|------|------|
| 1 | 1992 | 1 | Mali | Canada | A | N.r. | [44] |
| 2 | 2015 | 1 | USA | Australia | A | N.r. | [45] |

A, diagnosed by paired IgG IFA serology; N.r., none/not reported; Ref., reference; USA, United States of America.

HE or report case series of human infections with unusual *Ehrlichia* species, like *E. ruminantium* in South Africa [38], or *Candidatus* Ehrlichia erythraense in China [40] (Table 6 and paragraph below). Thus, it remains largely unclear if HME and to what extend HE exists outside North America, especially, in the absence of known transmission cycles, vector ticks or animal reservoirs. An unspecified febrile illness may easily be misdiagnosed as ehrlichiosis if based on serology only, as serological assays are hampered by cross-reactivity among Anaplasmataceae species, such as Anaplasma phagocytophilum [10]. This might explain some of the putative HE cases reported from regions outside the known ehrlichiosis endemic areas (Fig 5). The latter are well characterized for the US, with our analyzed data nicely mirroring the national surveillance data (Fig 6).

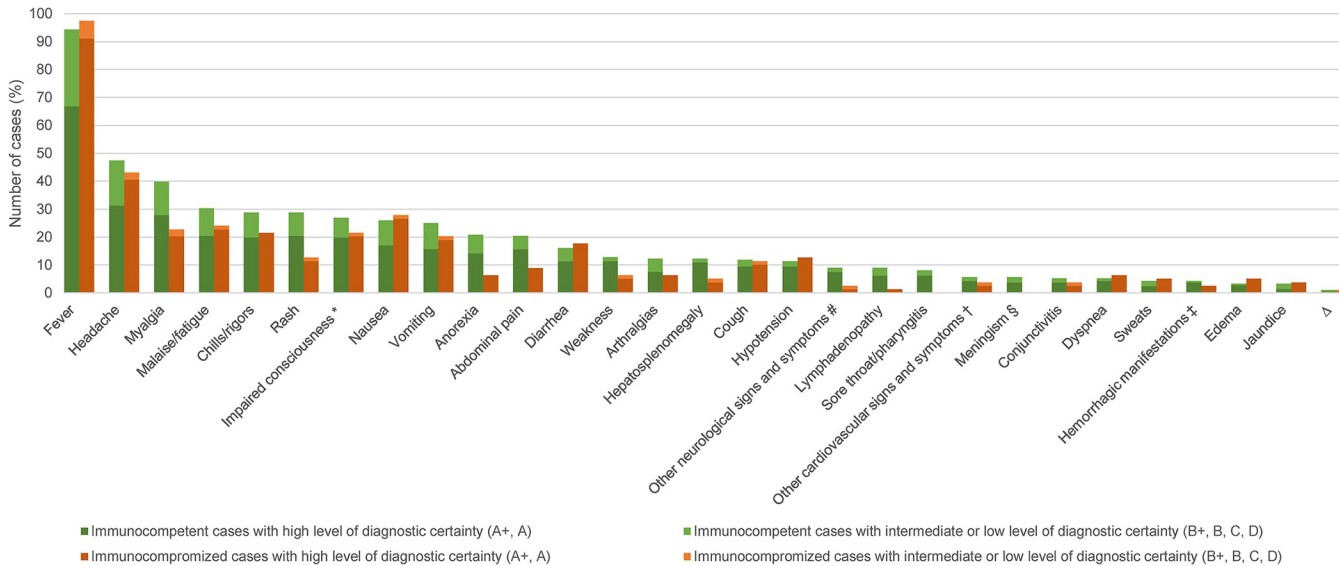

**Fig 8. Signs and symptoms of cases of human monocytotropic ehrlichiosis monoinfection reported with individual data (n = 290 cases: 211 immunocompetent, 79 immunocompromized).** A+, diagnosed by PCR, culture and/or immunostaining of biopsy/autopsy/tissue; A, diagnosed by paired IgG IFA serology; B+ diagnosed by microscopy of blood smear or by buffy coat preparation; B, diagnosed by single IgG IFA or ELISA serology; C, diagnosed by IgM serology; D, clinically diagnosed. * definition: altered mental state, confusion, somnolence, delirium, or coma. # other neurological symptoms included: ataxia (9.5%); dysarthria (9.5%); agitation (9.5%); hyperreflexia (9.5%); tingling in hand and feet (4.8%); glove and stocking numbness (4.8%); hallucinations (4.8%); hemiparesis (4.8%); hyperosmia (4.8%); tinnitus (4.8%); tremor (4.8%); decreased vision (4.8%); psychotic episodes (4.8%); visual changes + ataxia + tingling + areflexia + decreased vibration sensation with a glove and stocking pattern (4.8%); ataxia + distal sensory loss + global areflexia (4.8%); sixth cranial nerve palsy + diplopia + ataxia (4.8%); bilateral eye deviation + anisocoria (4.8%). † other cardiovascular signs and symptoms included: murmurs (46.7%); additional heart sounds (13.3%); atrial fibrillation (6.7%); hypertension (6.7%); bradycardia (6.7%); muffled heart sounds (6.7%); not further specified irregular heartbeat (6.7%); hypertension + murmur (6.7%). ‡ hemorrhagic manifestations: petechiae (45.5%); ecchymoses (18.1%); epistaxis (9.1%); subconjunctival hemorrhages (9.1%); gingival bleeding (9.1%); ecchymoses + subconjunctival hemorrhages (9.1%). § definition: headache plus neck stiffness and/or photophobia. Δ more rare signs and symptoms not included in the figure: <1–2%: dizziness, chest pain, otalgia, syncope, vertigo, epididymitis/epididymal pain, unspecified falls.

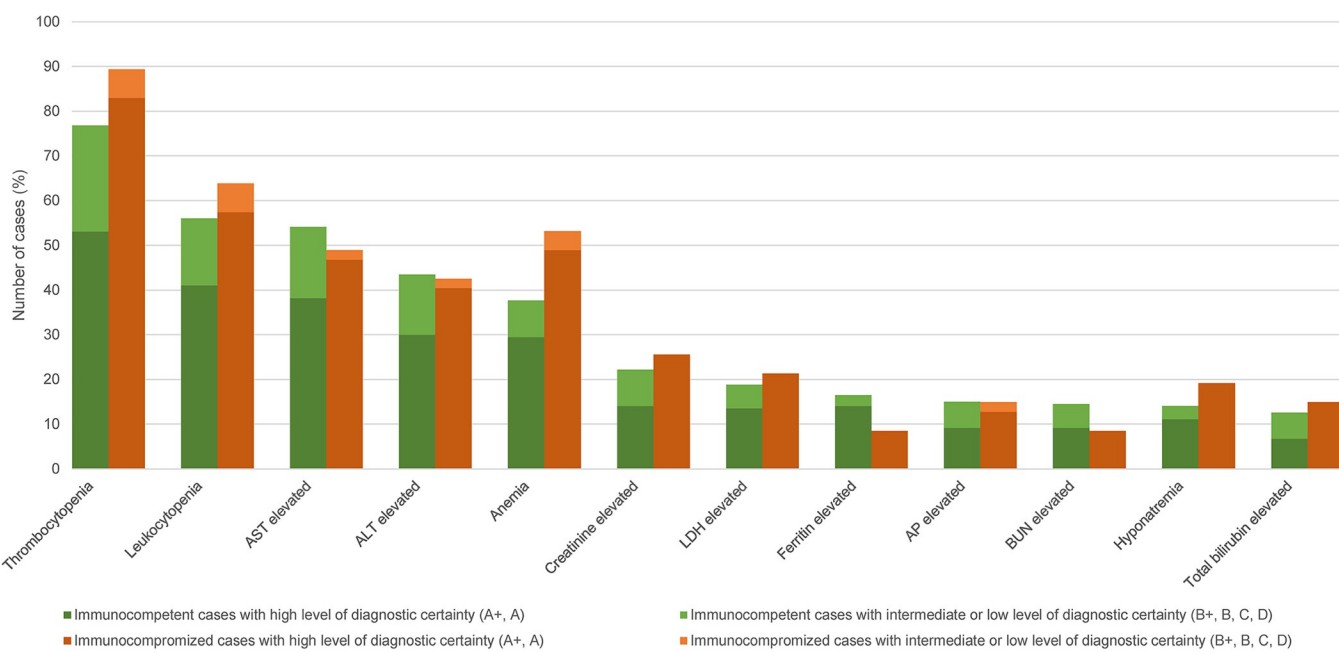

**Fig 9. Laboratory findings in human monocytotropic ehrlichiosis (n = 254 cases: 207 immunocompetent, 47 immunocompromized).** A+, diagnosed by PCR, culture and/or immunostaining of biopsy/autopsy/tissue; A, diagnosed by paired IgG IFA serology; B+ diagnosed by microscopy of blood smear or by buffy coat preparation; B, diagnosed by single IgG IFA or ELISA serology; C, diagnosed by IgM serology; D, clinically diagnosed. AST, aspartate aminotransferase; ALT, alanine aminotransferase; LDH, lactate dehydrogenase; AP, alkaline phosphatase; BUN, blood urea nitrogen [The cut-offs defining abnormal values of the individual laboratory parameters are listed in S7 Text].

## Non-canis/non-chaffeensis human Ehrlichiosis

Only a very small proportion of the published cases of HE are due to non-*canis*/non-*chaffeensis Ehrlichia* species (Table 6). The most frequently reported non-*canis*/non-*chaffeensis Ehrlichia* species in the US is *E. ewingii*, which accounts for 5.5% of reported HE cases, according to

**Table 13. Medians and ranges of laboratory parameters in human monocytotropic ehrlichiosis.**

| Laboratory parameters | Median | Range | Number of cases with available data (n) |
|---|---|---|---|
| Leukocytes ($10^3$/μl) | 2.9 | 0.4–28.9 | 177 |
| Thrombocytes ($10^3$/μl) | 62 | 6–413 | 191 |
| Hemoglobin (g/dl) | 11.1 | 4.8–16 | 106 |
| Hematocrit (%) | 36 | 0.14–0.46 | 48 |
| AST (U/l) | 189 | 6–5460 | 137 |
| ALT (U/l) | 124 | 9–1575 | 114 |
| AP (U/l) | 225 | 48–650 | 53 |
| Total bilirubin (μmol/l) | 28.1 | 1.4–1009 | 42 |
| LDH (U/l) | 945 | 212–19130 | 48 |
| BUN (mg/dl) | 49 | 12–136 | 35 |
| Creatinine (μmol/l) | 202 | 26–972 | 58 |
| Ferritin (μg/l) | 21187 | 402–85517 | 29 |
| Sodium (mmol/l) | 129 | 112–153 | 43 |

ALT, alanine aminotransferase; AP, alkaline phosphatase; AST, aspartate aminotransferase; BUN, blood urea nitrogen; LDH, lactate dehydrogenase.

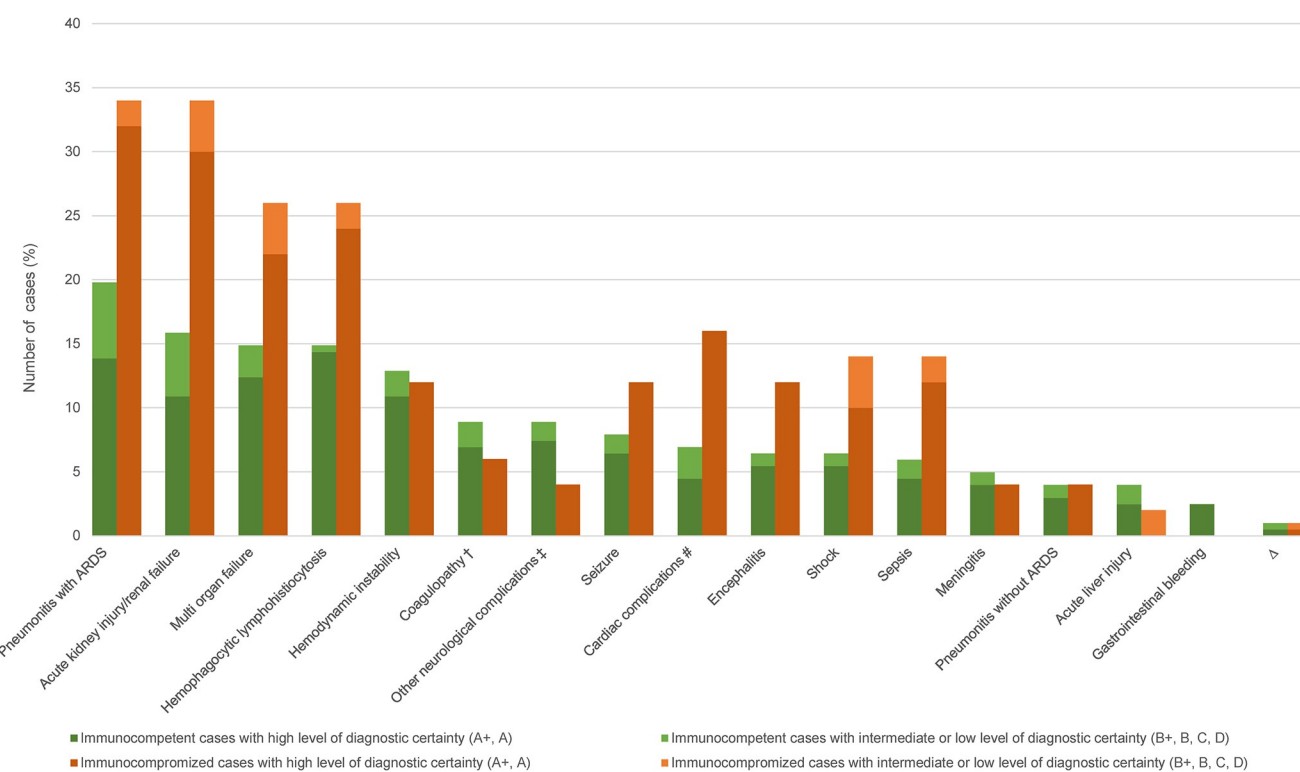

**Fig 10. Frequency of complications in human monocytotropic ehrlichiosis (n = 252: 202 immunocompetent, 50 immunocompromized)\*.** A+, diagnosed by PCR, culture and/or immunostaining of biopsy/autopsy/tissue; A, diagnosed by paired IgG IFA serology; B+ diagnosed by microscopy of blood smear or by buffy coat preparation; B, diagnosed by single IgG IFA or ELISA serology; C, diagnosed by IgM serology; D, clinically diagnosed. ARDS, acute respiratory distress syndrome. \* as in many cases, multiple complications were concomitantly present, the number of complications exceeds the number of cases. # cardiac complications: myocarditis (27.3%); cardiac arrest (22.7%); heart failure (18.2%); cardiac arrhythmia (13.6%); myocardial infarction (13.6%); ventricular tachycardia + heart failure (4.5%). † coagulopathy: disseminated intravascular coagulopathy (61.9%); unspecified coagulopathy (19.0%); disseminated intravascular coagulopathy + hemorrhagic manifestation (19.0%). ‡ other neurologic complications: coma (15.0%); cerebral edema with herniation (15.0%); Guillain-Barré syndrome (10.0%); hearing loss (10.0%); cranial nerve palsy (10.0%); acute inflammatory demyelinating polyradiculoneuropathy (5.0%); subarachnoid hemorrhage (5.0%); critical illness myopathy (5.0%); bilateral foot drop (5.0%); visual hallucinations (5.0%); psychotic episode (5.0%); partial Bell's palsy + monoparesis possibly due to a ischemic infarction (5.0%); sixth cranial nerve palsy + diplopia + ataxia (5.0%). Δ more rare complications not included in the figure: <1%: pulmonary edema, nephrotic syndrome, adrenal insufficiency, pulmonary embolism, rhabdomyolysis, appendicitis, cholestasis, abdominal compartement syndrome, ileus, syndrome of inappropriate antidiuretic hormone secretion, posterior uveitis, cholecystitis, graft failure with autologous reconstitution and residual acute myelogenous leukemia, spontaneous vaginal bleeding; esophageal ulcer + pulmonary hemorrhage, candiduria + severe thrombocytopenia requiring platelet transfusions, systemic hemorrhages + hemolysis, ecchymosis + thrombotic thrombocytopenic purpura-like illness, lower extremity deep vein thrombosis + *Candida spp.* fungemia, ascites + peripheral edema, mixed cryoglobulinemia + secondary membrano-proliferative glomerulonephritis + hand abscess, cytomegalie virus viremia + hemorrhagic cystitis, *Clostridium difficile* colitis + relapsed *Ehrlichia* infection, *Clostridium difficile* colitis + Candida albicans fungemia + *Pseudomonas aeruginosa* urinary tract infection + *Staphylococcus aureus* and *Candida albicans* sternal osteomyelitis, pulmonary embolism + endobronchial hemorrhage + cholestasis, pulmonary aspergillosis + cytomegalie virus infection + cholestasis + pancreatitis + intestinal infarction + congestive splenomegaly + gastric ulcers.

Heitmann and colleagues [10]. In our analysis we found that *E. ewingii* is reported in 3.3% of the published cases, but it should be pointed out that the proportion of reported *E. ewingii* cases is likely influenced by the used PCR assay [10].

In 2023, a new *Ehrlichia* species may have been discovered in China. In some provinces reported endemic typhus cases increased four-fold in spring 2021 compared to previous years. This led to an investigation to determine whether these cases were really all due to endemic typhus. The patients in question all tested negative for typhus group rickettsiae by PCR, but all serum samples showed IgM for *E. chaffeensis*. Subsequent phylogenetic analysis indicated that a new *Ehrlichia* species was the causative pathogen in these patients, who were misdiagnosed

**Table 14. Reported cases of human ehrlichiosis with hemophagocytic lymphohistiocytosis (n = 64).**

| No. | Year of publication | Age of patient (years) | Sex of patient | Pre-existing medical conditions | Immunosuppressive therapy | Level of diagnostic certainty regarding HE# | Co-infection | Fever >38.5°C | Splenomegaly | Cytopenia ≥2 lineages | Hemophago-cytosis in bone marrow | Hyperferritinemia | Low/absent NK-cell activity | Soluble CD25 >2400 U/ml | Hypofibrinogenemia or Hypertriglyceridemia | Diagnostic criteria HLH-2004 (x out of 8 criteria)* | Antimicrobial therapy | Complications other than HLH | Outcome | Ref. |
|---|---|---|---|---|---|---|---|---|---|---|---|---|---|---|---|---|---|---|---|---|
| 1 | 2010 | 10 | Male | N.r. | N.r. | A+ | N.r. | Yes | Yes | Yes | Yes | Yes | N.r. | N.r. | Yes | 6 | Doxy. | Seizure | Cured | [47] |
| 2 | 2011 | 10 | Female | N.r. | N.r. | A | N.r. | Yes | N.r. | N.r. | Yes | Yes | N.r. | Yes | Yes | 5 | Doxy. | Hemodynamic instability | Cured | [48] |
| 3 | 2011 | 13 | Male | N.r. | N.r. | A+ | N.r. | Yes | N.r. | Yes | Yes | Yes | N.r. | Yes | Yes | 6 | Doxy. | Seizure, hemodynamic instability | Cured | [48] |
| 4 | 2012 | 74 | Male | N.r. | N.r. | A+ | N.r. | Yes | N.r. | N.r. | Yes | Yes | N.r. | N.r. | Yes | 4 | Doxy. | N.r. | Cured | [49] |
| 5 | 2013 | 63 | Male | Kidney transplant | Yes | B | N.r. | Yes | Yes | Yes | N.r. | Yes | N.r. | N.r. | Yes | 5 | Doxy. | N.r. | Cured | [50] |
| 6 | 2014 | 52 | Female | N.r. | N.r. | A+ | N.r. | Yes | N.r. | Yes | Yes | Yes | N.r. | Yes | Yes | 6 | Doxy., Rif. | N.r. | Cured | [51, 52] |
| 7 | 2014 | 47 | Female | N.r. | N.r. | A+ | N.r. | Yes | N.r. | Yes | N.r. | Yes | N.r. | Yes | Yes | 5 | Doxy. | N.r. | Cured | [51, 52] |
| 8 | 2014 | 59 | Female | N.r. | N.r. | A+ | N.r. | Yes | N.r. | Yes | Yes | Yes | N.r. | N.r. | Yes | 5 | Doxy. | N.r. | Cured | [51, 52] |
| 9 | 2014 | 16 | Female | N.r. | N.r. | A+ | N.r. | Yes | N.r. | Yes | Yes | Yes | N.r. | N.r. | Yes | 5 | Doxy. | N.r. | Cured | [51, 52] |
| 10 | 2014 | 62 | Male | N.r. | N.r. | A+ | N.r. | Yes | Yes | Yes | N.r. | Yes | N.r. | N.r. | Yes | 5 | Doxy. | N.r. | Cured | [51, 52] |
| 11 | 2014 | 9 | Female | N.r. | N.r. | A+ | N.r. | Yes | N.r. | Yes | Yes | Yes | Yes | Yes | Yes | 7 | N.r. | N.r. | N.r. | [37] |
| 12 | 2014 | 7 | Female | N.r. | N.r. | A+ | N.r. | Yes | Yes | Yes | N.r. | Yes | Yes | Yes | Yes | 6 | N.r. | N.r. | N.r. | [37] |
| 13 | 2014 | 77 | Male | N.r. | N.r. | A+ | N.r. | Yes | N.r. | Yes | N.r. | Yes | Yes | Yes | Yes | 5 | N.r. | N.r. | N.r. | [37] |
| 14 | 2014 | 11 | Male | N.r. | N.r. | A+ | N.r. | Yes | N.r. | Yes | N.r. | Yes | Yes | Yes | Yes | 5 | N.r. | N.r. | N.r. | [37] |
| 15 | 2014 | 7 | Female | N.r. | N.r. | A+ | N.r. | Yes | N.r. | Yes | N.r. | Yes | Yes | Yes | Yes | 5 | N.r. | N.r. | N.r. | [37] |
| 16 | 2014 | 38 | Male | HIV+, Hepatitis C | N.r. | B | N.r. | N.r. | Yes | Yes | Yes | Yes | N.r. | N.r. | Yes | 5 | Doxy. | Septic shock, respiratory failure, AKI | Cured | [53] |
| 17 | 2015 | 7 | Male | N.r. | N.r. | A | N.r. | Yes | Yes | N.r. | Yes | Yes | Yes | N.r. | Yes | 5 | Doxy. | Hemodynamic instability | Cured | [54] |
| 18 | 2015 | 7 | Female | N.r. | N.r. | A+ | N.r. | Yes | N.r. | Yes | Yes | Yes | N.r. | N.r. | Yes | 4 | Doxy. | N.r. | Cured | [55] |
| 19 | 2015 | 9 | Male | N.r. | N.r. | A+ | EBV | Yes | N.r. | Yes | Yes | Yes | N.r. | Yes | Yes. | 6 | Doxy. | AKI, respiratory failure | Cured | [56] |
| 20 | 2016 | 41 | Female | Rheumatoid arthritis | Yes | B+ | N.r. | N.r. | N.r. | Yes | Yes | Yes | Yes | Yes | Yes | 5 | Doxy. | Meningoencephalitis, multi-organ failure | Cured | [57] |
| 21 | 2016 | 45 | Male | Ankylosing spondylitis | Yes | A+ | N.r. | N.r. | Yes | N.r. | Yes | Yes | Yes | N.r. | Yes | 4 | Doxy. | AKI, respiratory failure | Cured | [58] |
| 22 | 2017 | 66 | Female | HIV+ | N.r. | A+ | N.r. | N.r. | N.r. | N.r. | Yes | Yes | Yes | N.r. | Yes | 4 | N.r. | Septic shock | Death | [59] |
| 23 | 2018 | 26 | Male | DM | N.r. | A+ | N.r. | N.r. | Yes | N.r. | Yes | Yes | Yes | N.r. | Yes | 5 | Doxy. | DIC, sepsis, cerebral edema with hemorrhages and herniation | Death | [60] |
| 24 | 2018 | 70 | Male | Hepatitis C | N.r. | A+ | N.r. | Yes | N.r. | Yes | Yes | Yes | N.r. | N.r. | Yes | 5 | Doxy. | N.r. | Cured | [61] |
| 25 | 2018 | 28 | Male | N.r. | N.r. | B | EBV | Yes | N.r. | Yes | Yes | Yes | N.r. | N.r. | N.r. | 4 | Doxy. | N.r. | Cured | [62] |
| 26 | 2018 | 51 | Female | N.r. | N.r. | A+ | N.r. | N.r. | N.r. | Yes | Yes | Yes | N.r. | Yes | Yes | 5 | Doxy. | N.r. | Cured | [63] |
| 27 | 2018 | 62 | Female | Rheumatoid arthritis | Yes | B+ | N.r. | N.r. | Yes | N.r. | N.r. | Yes | N.r. | Yes | N.r. | 2 | Doxy. | Sepsis, ARDS, multi-organ failure | Death (unrelated) | [64] |
| 28 | 2019 | 45 | Male | Rheumatoid arthritis | Yes | A+ | N.r. | N.r. | Yes | N.r. | N.r. | Yes | Yes | Yes | Yes | 5 | Doxy. | Meningoencephalitis, AKI myocarditis, shock, ARDS | Cured | [65] |
| 29 | 2019 | 46 | Male | Rheumatoid arthritis | Yes | A+ | N.r. | N.r. | N.r. | Yes | Yes | N.r. | N.r. | N.r. | N.r. | 1 | Doxy. | Myocarditis, multi-organ failure | Cured | [66] |
| 30 | 2019 | 48 | Female | N.r. | N.r. | A+ | EBV | Yes | N.r. | Yes | Yes | N.r. | N.r. | N.r. | N.r. | 3 | Doxy. | Encephalopathy | Cured | [67] |
| 31 | 2019 | 47 | Male | HIV+ | N.r. | D | N.r. | N.r. | N.r. | Yes | N.r. | N.r. | N.r. | N.r. | N.r. | 1 | N.r. | Septic shock | Cured | [68] |
| 32 | 2020 | 8 | Female | N.r. | N.r. | A+ | N.r. | N.r. | Yes | N.r. | Yes | Yes | N.r. | N.r. | Yes | 4 | Doxy. | N.r. | Cured | [69] |
| 33 | 2020 | 3 | Female | N.r. | N.r. | A+ | N.r. | Yes | N.r. | N.r. | Yes | Yes | Yes | N.r. | Yes | 4 | Doxy. | AKI | Cured | [69] |
| 34 | 2020 | 7 | Female | N.r. | N.r. | A+ | N.r. | Yes | N.r. | N.r. | Yes | Yes | Yes | Yes | Yes | 4 | Doxy. | Shock, multi-organ failure | Death | [69] |
| 35 | 2020 | 13 | Female | N.r. | N.r. | A+ | N.r. | Yes | Yes | Yes | N.r. | Yes | Yes | N.r. | N.r. | 5 | Doxy. | Meningitis, seizures | Cured | [69] |
| 36 | 2020 | 15 | Male | N.r. | N.r. | A+ | N.r. | Yes | N.r. | Yes | Yes | Yes | N.r. | N.r. | Yes | 5 | Doxy. | Shock | Cured | [69] |
| 37 | 2020 | 7 | Female | N.r. | N.r. | A+ | N.r. | Yes | N.r. | Yes | N.r. | Yes | N.r. | N.r. | Yes | 5 | Doxy. | Shock, AKI, respir. failure, seizures | Cured | [69] |
| 38 | 2020 | 10 | Male | N.r. | N.r. | A+ | N.r. | Yes | N.r. | Yes | Yes | Yes | N.r. | Yes | Yes | 5 | Doxy. | Shock, AKI, respiratory failure, encephalopathy | Cured | [69] |

(Continued)

**Table 14.** (Continued)

| No. | Year of publication | Age of patient (years) | Sex of patient | Pre-existing medical conditions | Immunosuppressive therapy | Level of diagnostic certainty regarding HE# | Co-infection | Fever >38.5°C | Splenomegaly | Cytopenia ≥2 lineages | Hemophagocytosis in bone marrow | Hyperferritinemia | Low/absent NK-cell activity | Soluble CD25 >2400 U/ml | Hypofibrinogenemia or Hypertriglyceridemia | Diagnostic criteria HLH-2004 (x out of 8 criteria)* | Antimicrobial therapy | Complications other than HLH | Outcome | Ref. |
|---|---|---|---|---|---|---|---|---|---|---|---|---|---|---|---|---|---|---|---|---|
| 39 | 2020 | 9 | Female | N.r. | N.r. | A+ | N.r. | N.r. | Yes | N.r. | Yes | Yes | Yes | Yes | Yes | 6 | Doxy, Rif. | PRES, *Clostridioides difficile* colitis, relapsed ehrlichiosis | Cured | [69] |
| 40 | 2020 | 41 | Male | N.r. | N.r. | A+ | N.r. | Yes | N.r. | N.r. | N.r. | Yes | Yes | Yes | N.r. | 4 | Doxy. | Myocarditis | Cured | [70] |
| 41 | 2020 | 60 | Female | N.r. | N.r. | A+ | N.r. | Yes | N.r. | Yes | N.r. | Yes | Yes | N.r. | Yes | 5 | Doxy. | Myocarditis, encephalitis, multi-organ failure | Death | [70] |
| 42 | 2020 | 68 | Male | N.r. | N.r. | A+ | N.r. | Yes | N.r. | Yes | N.r. | Yes | Yes | Yes | Yes | 6 | Doxy. | Myocarditis, respiratory failure, shock | Death | [70] |
| 43 | 2020 | 63 | Male | Coronary heart dis. | N.r. | A+ | N.r. | Yes | Yes | N.r. | N.r. | Yes | N.r. | Yes | Yes | 5 | Doxy. | Respiratory failure, AKI | Cured | [71] |
| 44 | 2021 | 66 | Male | Kidney transplant, Still disease | Yes | A+ | N.r. | N.r. | N.r. | N.r. | Yes | Yes | N.r. | N.r. | Yes | 3 | N.r. | Meningoencephalitis, respiratory failure | Death | [72] |
| 45 | 2021 | 72 | Male | Multiple myeloma, ASCT | Yes | A+ | N.r. | N.r. | N.r. | Yes | N.r. | Yes | Yes | Yes | Yes | 5 | Doxy. | AKI, respiratory failure | Death | [73] |
| 46 | 2021 | 70 | Male | Kidney transplant | Yes | A+ | N.r. | N.r. | N.r. | Yes | Yes | N.r. | N.r. | N.r. | N.r. | 2 | Doxy. | Seizure, sepsis | Death | [17] |
| 47 | 2021 | 66 | Male | Kidney transplant, Still's disease | Yes | A+ | N.r. | Yes | N.r. | Yes | Yes | N.r. | N.r. | N.r. | N.r. | 3 | N.r. | Encephalopathy, AKI, cardiac arrest | Death | [17] |
| 48 | 2021 | 6 | Male | Kidney transplant | Yes | A+ | N.r. | Yes | N.r. | N.r. | Yes | Yes | N.r. | N.r. | N.r. | 3 | Doxy. | N.r. | Cured | [17] |
| 49 | 2021 | 9 | Male | N.r. | N.r. | A+ | N.r. | N.r. | Yes | Yes | N.r. | Yes | N.r. | N.r. | Yes | 4 | Doxy. | N.r. | Cured | [74] |
| 50 | 2021 | 49 | Male | Ankylosing spondylitis, psoriatic arthritis | Yes | A+ | RMSF | Yes | Yes | N.r. | Yes | Yes | Yes | Yes | Yes | 7 | Doxy. | Respiratory failure, AKI | Cured | [75] |
| 51 | 2021 | 62 | Female | N.r. | N.r. | A+ | N.r. | Yes | N.r. | N.r. | Yes | Yes | N.r. | Yes | N.r. | 4 | Doxy. | Hypotension, respiratory failure | Cured | [75] |
| 52 | 2021 | 81 | Female | Rheumatoid arthritis | N.r. | A+ | EBV | N.r. | N.r. | N.r. | Yes | N.r. | N.r. | Yes | Yes | 5 | Doxy. | Hemodynamic collapse | Death | [76] |
| 53 | 2021 | 59 | Male | DM | N.r. | A+ | N.r. | Yes | Yes | N.r. | N.r. | Yes | N.r. | N.r. | Yes | 4 | Doxy. | N.r. | Cured | [77] |
| 54 | 2021 | 16 | Male | N.r. | N.r. | A+ | N.r. | Yes | Yes | N.r. | N.r. | Yes | N.r. | N.r. | Yes | 4 | Doxy. | Shock | Cured | [78] |
| 55 | 2022 | 80 | Female | Aortic valve craft | N.r. | B | N.r. | N.r. | Yes | Yes | N.r. | Yes | N.r. | Yes | Yes | 3 | Doxy. | Multi-organ failure | Death | [79] |
| 56 | 2022 | 70 | Male | N.r. | Yes | A+ | N.r. | N.r. | N.r. | N.r. | Yes | Yes | N.r. | Yes | N.r. | 3 | Doxy. | Encephalopathy, multi-organ failure, endocarditis | Death | [79] |
| 57 | 2022 | 16 | Male | Kidney transplant | Yes | A+ | N.r. | Yes | N.r. | N.r. | N.r. | Yes | N.r. | Yes | Yes | 5 | Doxy. | AKI | Cured | [80] |
| 58 | 2022 | 61 | Female | Liver transplant | Yes | A+ | N.r. | Yes | N.r. | N.r. | N.r. | N.r. | N.r. | N.r. | N.r. | 1 | Doxy. | Encephalopathy, AKI, cardiac arrest | Sequelae | [81] |
| 59 | 2022 | 59 | Male | MGUS | N.r. | A+ | N.r. | Yes | N.r. | N.r. | Yes | Yes | N.r. | N.r. | N.r. | 3 | Doxy. | N.r. | Cured | [82] |
| 60 | 2023 | 10 | Male | N.r. | N.r. | A+ | N.r. | Yes | Yes | Yes | N.r. | Yes | N.r. | Yes | N.r. | 5 | Doxy. | N.r. | Cured | [83] |
| 61 | 2023 | 4 | Female | N.r. | N.r. | A+ | N.r. | Yes | Yes | N.r. | N.r. | Yes | N.r. | Yes | Yes | 5 | Doxy. | N.r. | Cured | [84] |
| 62 | 2023 | 67 | Female | Autoimmune hepatitis, cirrhosis | Yes | A+ | N.r. | N.r. | N.r. | Yes | Yes | Yes | N.r. | N.r. | Yes | 3 | Doxy. | Septic shock | Sequelae | [84] |
| 63 | 2023 | 69 | Female | N.r. | N.r. | A+ | N.r. | N.r. | N.r. | Yes | Yes | Yes | N.r. | Yes | Yes | 5 | Doxy. | Myocarditis, pneumonia, AKI | Cured | [84] |
| 64 | 2023 | 47 | Male | N.r. | N.r. | A+ | N.r. | Yes | Yes | N.r. | N.r. | Yes | N.r. | N.r. | Yes | 5 | Doxy. | Multi-organ failure | Cured | [84] |

AKI, acute kidney injury; ARDS, adult respiratory distress syndrome; ASCT, autologous stem cell transplantation; CMV, cytomegalie virus; DIC, disseminated intravascular coagulopathy; DM, diabetes mellitus; Doxy, Doxycycline; ESRD, end-stage renal disease; HIV+, human immunodeficiency virus positive; HLH, hemophagocytic lymphohistiocytosis; IgG, immunoglobulin G; MGUS, monoclonal gammopathy of undetermined significance; NK cell, natural killer cell; N.r. none/not reported; PRES, posterior reversible encephalopathy syndrome; Ref, reference; Rif., Rifampicin.

* HLH-critera 2004 [85]

# A, diagnosed by paired IgG IFA serology; A+, diagnosed by PCR, immunostaining of biopsy/autopsy tissue and/or culture; B, diagnosed by IgG single titre IFA serology; B+, diagnosed by microscopy; D, diagnosed clinically.

as endemic typhus. The name *Candidatus* Ehrlichia erythraense was proposed as all 19 patients presented with a rash [40].

## Coinfections

Among the reviewed HE cases, we found a coinfection rate with other tick-borne pathogens of 2.1% (26/1260) (Table 7). However, reported data on the rate of co-infection in HE cases should generally be interpreted with caution (limited sample sizes, questionable representativeness, lack of systematic testing for coinfections). In addition, the report of concomitantly diagnosed coinfections should especially interpreted with caution if the causative pathogens are exclusively transmitted by different tick species.

## Diagnostic

The most frequently reported diagnostic methods used to diagnose HE are PCR and serology (Table 8). PCR is highly specific but specificity varies among assays. In the reviewed publications, often an unspecific *Ehrlichia* PCR assay was used, which might lead to an overestimation of HE cases, as some of these assays not only detect *Ehrlichia* spp. but also *Anaplasma* spp. [117,118]. In serology, similar specificity problems arise due to the cross-reactivity of serological assays with pathogens from the Anaplasmataceae family as well as between different *Ehrlichia* species [21]. In addition, false positive serological results may also be seen. A notable example in this regard is that of a woman with clinical symptoms consistent with HE who had a positive *E. chaffeensis*-specific IgM and IgG test result but did not respond to appropriate antimicrobial treatment. This patient was eventually diagnosed with systemic lupus erythematosus (SLE), and after SLE-specific treatment, her ehrlichiosis serology became negative, confirming that her false-positive test was likely due to SLE [116].

The higher proportion of CRID diagnosed by PCR compared to CRNID (Table 8) is probably due to the fact that CRID were more frequently reported in context of acute infection, for which PCR is the ideal tool, whereas CRNID were more frequently reported in retrospective studies and therefore diagnosed retrospectively by serology. Microscopy of blood smears for the presence of morulae is rarely performed. This is probably due to the fact that the presence of morulae in HME is rare overall (much rarer than in anaplasmosis) and therefore the sensitivity is low, and that PCR is a much more sensitive method for acute diagnosis. In connection with microscopy, it should be noted that the tropism of *E. chaffeensis* for monocytes is not 100%, contrary to the suggestive designation human *monocytotropic* ehrlichiosis (Table 9).

Immunostaining of biopsy tissue and culture is rarely performed (Table 8), which is understandable, considering that these diagnostic methods are technically challenging as well as resource and time demanding and rarely available.

## Patients' characteristics

HME was more commonly diagnosed in males than in females (M:F ratio = 1.84) and mostly seen in middle-aged and older individuals, which corresponds to US surveillance data [119] and most likely reflects the higher recreational and/or occupational exposure risk of males to ticks. Of the cases we analyzed, 26.7% were reported in immunocompromized patients. In 54.9% of these patients the immunocompromization was due to past organ transplantation. This is in complete agreement with the literature, which usually states that a quarter of HME patients fall into the group of immunocompromized patients [8]. HME is therefore an opportunistic infection that should be included in the differential diagnoses of immunocompromized patients, particularly organ transplant recipients, living in endemic areas.

Table 15. Antimicrobial treatment of human ehrlichiosis (n = 356) considered effective*.

| Antimicrobial treatment | Number of cases n (%) |
|---|---|
| Doxycycline | 298 (83.7) |
| Tetracycline | 21 (5.9) |
| Chloramphenicol | 20 (5.6) |
| Minocycline | 4 (1.1) |
| Rifampicin | 2 (<1) |
| Multiple compounds | 11 (3.1) |
| Doxycycline + chloramphenicol | 8 |
| Doxycycline + rifampicin | 2 |
| Tetracycline + chloramphenicol | 1 |

* tetracyclines (doxycycline, minocycline) and rifampicin were considered effective/appropriate; chloramphenicol was also considered effective/appropriate, although evidence is limited; beta-lactams, quinolons, macrolides, aminoglycosides, glycopeptides, nitroimidazoles, sulphonamides, and lincosamide were considered ineffective/inappropriate.

Among our reviewed cases, the hospitalization rate was 83.2%, which is considerably higher than the 57% reported in US national surveillance data [10], but similar to the hospitalization rate of 88.4% reported by a tertiary care centre in an endemic area [116]. The discrepancy in the reported hospitalization rates is thus plausibly explained by the nature of the underlying data sources with our data primarily reflecting the more likely published data on severe cases.

## Route of transmission

The most common route of transmission of HME is by tick bite, which is confirmed by the fact that in 96.1% of the reviewed HME cases the reported suspected route of transmission was

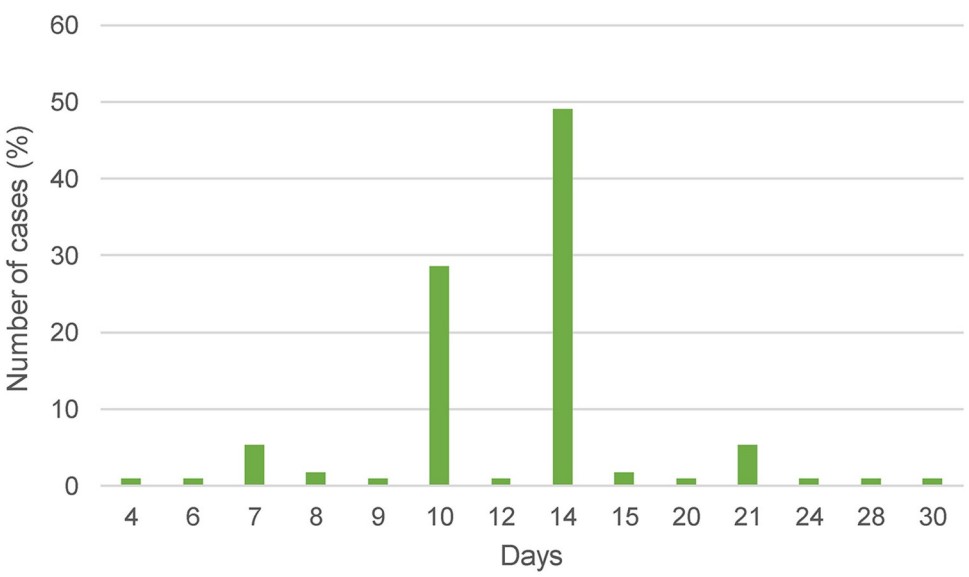

Fig 11. Duration of antimicrobial treatment for human ehrlichiosis (n = 112).

Table 16. Reported fatal cases of human monocytotropic ehrlichiosis (n = 34).

| No. | Year of publication | Age of patient (years) | Sex of patient | Country of infection | Pre-existing medical conditions | Immunosuppressive therapy | Time between first symptoms and medical presentation (days) | Level of diagnostic certainty # | Antimicrobial treatment | Time between presentation to hospital and specific therapy (days) | Complications/cause of death | Time from first symptoms to death | Ref. |
|---|---|---|---|---|---|---|---|---|---|---|---|---|---|
| 1 | 1990 | 68 | M | USA | Hypertension | N.r. | N.r. | A | Doxy., chloramphenicol | N.r. | AKI | 68 | [86] |
| 2 | 1991 | 67 | M | USA | N.r | N.r. | N.r. | B | Chloramphenicol | N.r. | Multi organ failure | N.r. | [87] |
| 3 | 1993 | 68 | M | USA | Lobectomy | N.r. | 7 | B | Doxy., chloramphenicol | 7 | Multi organ failure | 68 | [88] |
| 4 | 1993 | 41 | F | USA | HIV+, intestinal strongyloidiasis | N.r. | N.r. | A+ | N.r. | N.r. | Respiratory failure, pulmonary hemorrhage | 16 | [89] |
| 5 | 1996 | 36 | M | USA | HIV+ | N.r. | N.r. | A | N.r. | N.r. | AKI | N.r. | [90] |
| 6 | 1996 | 73 | M | USA | N.r. | N.r. | N.r. | B | Doxy. | N.r. | AKI | N.r. | [91] |
| 7 | 1997 | 13 | F | USA | N.r. | N.r. | 7 | B | N.r. | N.r. | Encephalopathy, multi-organ failure | 9 | [92] |
| 8 | 1997 | 51 | F | USA | Hashimoto's thyreoiditis | N.r. | 7 | A | Doxy. | 5 | Encephalopathy, multi-organ failure | 13 | [93] |
| 9 | 1999 | 22 | M | USA | N.r. | N.r. | 7 | A+ | Doxy. | 0 | Seizures, multi organ failure | 10 | [94] |
| 10 | 1999 | 38 | M | USA | AIDS, Hepatitis B, Hepatitis C | N.r. | N.r. | A+ | Unspecified appropriate agent | N.r. | Seizures, multiple-organ failure | N.r. | [94] |
| 11 | 2001 | 52 | M | USA | HIV+ | N.r. | N.r. | A+ | N.r. | N.r. | AKI, pneumonia | 10 | [95] |
| 12 | 2001 | 66 | M | USA | N.r. | N.r. | 10 | A+ | N.r. | N.r. | Seizures, multi organ failure | 18 | [95] |
| 13 | 2001 | 80 | F | USA | MGUS | N.r. | 6 | A+ | Doxy. | N.r. | Respiratory failure | 7 | [95] |
| 14 | 2001 | 20 | M | Brazil | N.r. | N.r. | N.r. | B | N.r. | N.r. | Respiratory insufficiency, AKI | 10 | [95] |
| 15 | 2004 | 19 | F | USA | N.r. | N.r. | 7 | B | Doxy. | N.r. | Multi organ failure | N.r. | [96] |
| 16 | 2006 | 52 | M | USA | HIV+ | N.r. | 4 | A+ | Doxy. | N.r. | Pneumonia, AKI | 10 | [42] |
| 17 | 2006 | 15 | M | Brazil | N.r. | Yes | N.r. | A | Chloramphenicol | N.r. | Encephalitis | N.r. | [97] |
| 18 | 2009 | 40 | M | USA | N.r. | N.r. | 14 | A+ | N.r. | N.r. | Encephalopathy, multi-organ failure | N.r. | [98] |
| 19 | 2010 | 7 | M | USA | ALL | Yes | 1 | A+ | Doxy. | N.r. | Respiratory failure, seizure, cardiac arrest | 9 | [99] |
| 20 | 2011 | 60 | F | USA | Melanoma | N.r. | 14 | A+ | N.r. | N.r. | Coagulopathy, multi organ failure | 16 | [100] |
| 21 | 2012 | 7 | M | USA | ALL | Yes | 4 | A+ | Doxy. | N.r. | Septic shock, Multi organ failure, | 10 | [101] |
| 22 | 2014 | 56 | F | USA | N.r. | N.r. | N.r. | A+ | Doxy. | N.r. | Seizure, myocardial infarction, multi-organ failure | N.r. | [102] |
| 23 | 2014 | 45 | M | USA | N.r. | N.r. | N.r. | A+ | N.r. | N.r. | Myocardial infarction, status epilepticus, meningitis, multi organ failure | N.r. | [102] |

*(Continued)*

**Table 16.** (Continued)

| No. | Year of publication | Age of patient (years) | Sex of patient | Country of infection | Pre-existing medical conditions | Immunosuppressive therapy | Time between first symptoms and medical presentation (days) | Level of diagnostic certainty # | Antimicrobial treatment | Time between presentation to hospital and specific therapy (days) | Complications/cause of death | Time from first symptoms to death | Ref. |
|---|---|---|---|---|---|---|---|---|---|---|---|---|---|
| 24 | 2016 | 31 | F | Mexico | N.r. | N.r. | 15 | A+ | Doxy. | N.r. | Encephalopathy, multi organ failure | 27 | [103] |
| 25 | 2017 | 66 | F | USA | HIV+ | N.r. | N.r. | A+ | N.r. | N.r. | Septic shock, HLH | N.r. | [59] |
| 26 | 2017 | 45 | M | USA | N.r. | N.r. | 7 | A+ | Doxy. | N.r. | Abdominal compartment syndrome, multi-organ failure | 14 | [104] |
| 27 | 2018 | 26 | M | USA | DM | N.r. | N.r. | A+ | Doxy. | N.r. | HLH, DIC, sepsis, cerebral edema, focal hemorrhages and herniation | N.r. | [60] |
| 28 | 2020 | 37 | M | Mexico | Femoral fracture | N.r. | N.r. | A+ | N.r. | N.r. | Septic shock | N.r. | [16] |
| 29 | 2020 | 95 | F | USA | Dementia | N.r. | N.r. | A+ | Doxy. | N.r. | Encephalopathy | N.r. | [105] |
| 30 | 2020 | 7 | F | USA | N.r. | N.r. | 7 | A+ | Doxy. | N.r. | HLH, multi organ failure | N.r. | [69] |
| 31 | 2021 | 66 | M | USA | Adult Still disease | Yes | 7 | A+ | N.r. | N.r. | HLH, respiratory failure, meningoencephalitis | 28 | [72] |
| 32 | 2021 | 72 | M | USA | Multiple myeloma, ASCT | Yes | N.r. | A+ | Doxy. | 4 | HLH, multi-organ failure | N.r. | [73] |
| 33 | 2021 | 70 | M | USA | Kidney transplant | Yes | N.r. | A+ | Doxy. | N.r. | HLH, seizure, sepsis | 34 | |
| 34 | 2021 | 66 | M | USA | Kidney transplant, Adult Still disease | Yes | 1 | A+ | N.r. | N.r. | HLH, cardiac arrest | 12 | [17] |

AIDS, acquired immune deficiency syndrome; AKI, acute kidney injury/kidney failure; ARDS, acute respiratory distress syndrome; ASCT, autologous stem cell transplantation; DIC, disseminated intravascular coagulopathy; DM, diabetes mellitus; Doxy, Doxycycline; ESRD, end-stage renal disease; HIV+, human immunodeficiency virus positive; HLH, hemophagocytic lymphohistiocytosis; IgG, immunoglobulin G;MGUS, monoclonal gammopathy of undetermined significance; N.r.; None/not reported; UTI, urinary tract infection; USA, United States of America.
# A, diagnosed by paired IgG IFA serology; A+, diagnosed by PCR, immunostaining of biopsy/autopsy tissue and/or culture; B, diagnosed by IgG single titre IFA serology.

Table 17. Reported sequelae of human monocytotropic ehrlichiosis (n = 11).

| No. | Year of publication | Age of patient (years) | Sex of patient | Country of infection | Pre-existing medical conditions | Immuno-suppressive treatment | Time between first symptoms and presentation to hospital/physician (days) | Antimicrobial treatment | Time between presentation to hospital and specific therapy (days) | Complications | Sequelae | Ref. |
|---|---|---|---|---|---|---|---|---|---|---|---|---|
| 1 | 1990 | 35 | Male | USA | N.r. | N.r. | 2 | Doxycycline | N.r. | N.r. | Failed to regain his previous vigor, short-term memory impairment, recurrent parietal headaches. | [106] |
| 2 | 1990 | 66 | Male | USA | N.r. | N.r. | 7 | Tetracycline | 0 | N.r. | Persistent weakness and aching across the chest and shoulders for two months. | [107] |
| 3 | 1992 | 63 | Female | USA | N.r. | N.r. | N.r. | Doxycycline | N.r. | Hemodynamic instability | Fine tremor of her upper extremities, anorexia and weakness that resolved after four weeks. | [108] |
| 4 | 1992 | 9 | Female | USA | N.r. | N.r. | 14 | Chloramphenicol | 3 | DIC | Muscle strength was diminished for eight weeks. | [109] |
| 5 | 1998 | 47 | Female | USA | N.r. | N.r. | 10 | Doxycycline | N.r. | Sepsis, gastrointestinal hemorrhage, seizure, multi-organ failure, pulmonary embolism, | Residual neurologic deficits including brachial plexopathy, autonomic dysfunction and memory impairment. | [110] |
| 6 | 2000 | 7 | Female | USA | N.r. | N.r. | 7 | Chloramphenicol | N.r | Hemodynamic instability, ARDS, ascites, peripheral edema, DIC | Residual neurologic signs including ataxia, lower limb clonus, symmetrical hand tremor, photophobia, cognitive deficits. Motor and reflex signs resolved within 1 week, cognitive and affect impairment persisted for 3 weeks. | [111] |
| 7 | 2013 | 73 | Female | USA | N.r. | N.r. | 4 | Doxycycline | N.r. | Myocarditis | Cardiac decompensation resolved over several days, ejection fraction of 70% on echocardiogram 10 weeks after admission. | [112] |
| 8 | 2017 | 16 | Male | Colombia | N.r. | N.r. | N.r. | Doxycycline | N.r. | Seizures | Seizures | [113] |

(Continued)

**Table 17.** (Continued)

| No. | Year of publication | Age of patient (years) | Sex of patient | Country of infection | Pre-existing medical conditions | Immuno-suppressive treatment | Time between first symptoms and presentation to hospital/physician (days) | Antimicrobial treatment | Time between presentation to hospital and specific therapy (days) | Complications | Sequelae | Ref. |
|-----|------|------|------|------|------|------|------|------|------|------|------|------|
| 9 | 2021 | 15 | Male | USA | N.r. | N.r. | 4 | Doxycycline | N.r. | Meningoencephalitis, sepsis | Increased agitation, impulsivity and poor cognition persisted 45 days after onset of illness. | [114] |
| 10 | 2022 | 61 | Female | USA | Liver transplantation | Yes | 14 | Doxycycline | N.r. | HLH, encephalitis, AKI, cardiac arrest | Temporary outpatient dialysis required. | [81] |
| 11 | 2023 | 67 | Female | USA | Autoimmune hepatitis, liver cirrhosis | Yes | 7 | Doxycycline | 0 | HLH, septic shock | Relapse of severe thrombocytopenia a few weeks after admission. | [84] |

ARDS, acute respiratory distress syndrome; AKI, acute kidney injury/kidney failure; DIC, disseminated intravascular coagulopathy; HLH, hemophagocytic lymphohistiocytosis; N.r., None/not reported; USA, United States of America.

a tick bite. Transmission through blood transfusions (n = 3; Table 10) and organ transplants (n = 7; Table 11) is reported, but appears to be rather rare overall.

Whether HE is vertically transmissible in case of infection during pregnancy remains unclear. We encountered only one case report of HE during pregnancy, and in this case no vertical transmission was reported.

### Travel-related cases

With only two ever reported cases, HE appears to not be a relevant infection in the context of international travel. Although the awareness of physicians to suspect HE as well as the diagnostic capacities to confirm HE is likely very low outside US endemic areas, we do not consider it likely that HE is frequent among international travelers.

### Signs and symptoms

HME presents similarly to many other febrile illnesses without specific signs and symptoms (Fig 8) and can be easily misdiagnosed or overlooked as it is often mild, oligosymptomatic and self-limiting, even in the absence of antimicrobial treatment. The majority of ehrlichiosis infections may even be asymptomatic and therefore go unnoticed, as supported by respective case reports [114]. Since Ehrlichiae belong to the order Rickettsiae, it is not surprising that the classic clinical triad of fever, headache and skin rash is also observed in HE. However, we found this classical triad reported in only 10% of the reviewed cases. The presence of a rash was described in 24.5% of the reviewed cases, with some discrepancies when looking on certain subgroups: e.g. rash was reported in 43% of pediatric and 19% of adults HME cases and in 28% of immunocompetent and 13% of immunocompromized cases, respectively. The range and difference in the age-specific proportions is roughly in line with the literature, where the presence of a rash is described in approximately 48–60% of pediatric and <30% of adult HME cases [42,120]. Regarding our finding of a lower, but still considerable rate of rash in immunocompromised patients, it is interesting that in a retrospective study comparing immunocompetent HME cases (n = 43) with organ transplanted HME cases (n = 15), the only statistically significant different symptom was rash, being present in 38% of immunocompetent but 0% of organ transplant patients [31].

### Laboratory findings

Laboratory findings in HME cases are not specific, but often characterized by the combination of cytopenia (with thrombocytopenia being the most common, followed by leukopenia and less commonly mild non-hemolytic anemia) and elevated transaminase levels (Fig 9). However, this pattern may also be observed with other infections, including many other tick-borne infections (e.g., anaplasmosis, rickettsiosis, tularemia, babesiosis, tick-borne relapsing fever, Q fever, and tick-borne arboviral infections) [121].

### Complications

The reported rate of complications among the reviewed monoinfected CRID was 63.1% and thus rather high when compared to the complication rate of up 17% according to the literature [25,115,116]. This difference is very likely explained by reporting and publication bias, but very likely also by differences in case definition. We defined a complicated course of HE as a case whose severity requires specific therapeutic interventions/care beyond the administration of antimicrobial therapy (e.g., respiratory support, renal replacement therapy, circulatory support, blood transfusion, intensive care monitoring etc.), whereas many publications restricted

the definition of complications to life-threatening organ-failure. Also, the results of our analysis regarding complications very likely reflect a worst-case scenario, as the denominator used to calculate the incidence of complications was conservatively chosen when using only data from cases in which data on complications where available. In addition, 25% of the cases with reported complications we reviewed were immunocompromised. As complications are known to occur more frequently in immunocompromised individuals [122–125], this may also have contributed to the higher complication rate we found.

The most common complications were ARDS, acute renal failure, multi-organ failure and HLH, with the first two complications occurring statistically significantly more frequently in immunocompromized patients (Fig 10). Particularly HLH appears to be a prominent complication of HE. HLH comes in two forms, primary (inherited) HLH, and secondary (acquired) HLH (sHLH). The latter has many known triggers, including neoplasms, autoimmune processes, and particularly infections, with the most common infectious cause being viral [85,126]. sHLH results from defective hyperactivation of natural killer and cytotoxic T-cells that can progress to multi-organ failure [126]. In septic patients, the incidence of sHLH is estimated between 3.7 and 4.3% [126]. Among the reviewed cases with reported data on complications, HLH was reported in 17% (43/252) of the cases. This number is very likely overestimating the frequency of HE-related HLH due to reporting and publication bias, but even if assuming the absence of HLH in all reviewed HE-cases not specifically reporting the absence of HLH, the incidence of HLH would still be 6.5% (64/980). Although many pathogens are known as possible triggers of HLH, we have not found any other pathogen in the literature that has a similarly high association rate with HLH as *Ehrlichia*. We found a CFR among HE-related HLH cases of 18% (12/64). Since the overall reported case fatality rate of HLH is 50–80% [127], HE-related HLH appears to have a comparatively good outcome.

## Treatment

All patients with a suspected diagnosis of HME should be treated empirically without delay, regardless of whether the diagnosis is confirmed, as delayed treatment leads to prolonged illness, longer hospitalization, and an increased risk of complications [128]. Doxycycline is the most commonly and widely used antimicrobial agent (Table 16) and the recommended treatment of choice. The most frequently recommended and most widely adopted dosing regimen for adults is 100mg of doxycycline twice daily [26], but no studies have yet been conducted to define an optimal treatment regimen for HME. In our analysis, the median duration of antimicrobial treatment was 14 days, with a clear tendency of physicians to either opt for a 7-, 10-, 14- or 21-day treatment regimens (Fig 11). The tendency to opt for 10–14 days is probably due to the fact that physicians want possible Lyme co-infection to be covered [42,129], while for HME, a treatment duration of 4–5 days (corresponding to three days after defeverence) is otherwise considered adequate [26,130]. Our analysis confirms that fever resolves rapidly (mostly within 2 day) after starting appropriate antimicrobial treatment. Thus, in the case of persisting fever, the diagnosis of HME should be reconsidered. While tetracycline is contraindicated in children <8 years of age due to the risk of irreversible dental staining, administration of doxycycline for up to 21 days is considered safe at any age [131–133]. The recommended pediatric dose of doxycycline for children weighing less than 45 kg is 2.2mg/kg body weight twice daily [26]. Regarding pregnant woman, doxycycline has shown no evidence for increased teratogenicity [133] and the US CDC states: "in cases of life-threatening allergies to doxycycline, severe doxycycline intolerance, and in some pregnant patients for whom the clinical course of ehrlichiosis appears mild, physicians might consider alternate antibiotics" [130].Of note, with regard to severe

doxycycline allergies, a protocol for rapid desensitization to doxycycline has been published [134]. Besides doxycycline (or tetracycline), rifampin is the only potentially effective alternative, although there is little data available on it. Rifampin appears effective against *E. chaffeensis* in vitro [135] whereas animal model data on the efficacy of rifampicin against *E. canis* is inconsistent [136]. The successful use of rifampin in HGA is reported [137, 138], but respective reports on its use in children and pregnant women with HME are lacking. Only one case report has been published describing a successful use of rifampin for treatment of HME [139]. In addition, rifampin would not be effective in the case of Rocky Mountain Spotted Fever (RMSF), a feared differential diagnosis of ehrlichiosis. The latter is particularly relevant as their endemic areas in the USA overlap considerably. In vitro data show that chloramphenicol, ciprofloxacin, cotrimoxazole (trimethoprim/sulfamethoxazole), erythromycin, penicillin, and gentamicin are not effective against *E. chaffeensis* [135].

## Outcome

Most patients recover without sequelae. In only 4% of cases with data on outcome, sequelae were reported, mostly lasting for a few weeks after acute disease (Table 17). The overall case fatality rate in our analysis was 11.6%, with a higher CFR in immunocompromised than in immunocompetent patients (16.5% vs 9.9%). In the literature, the case fatality rate of ehrlichiosis is much lower. The highest case fatality rate is seen in children <4 years and in patients >70 years with 4% and 3%, respectively [10]. The higher case fatality rate in our analysis is likely explained by publication bias as cases with complications and/or fatal outcome are much more likely to be published than mild or asymptomatic cases (Table 16). Regarding immunity after infection with *Ehrlichia*, there is no conclusive evidence whether patients are susceptible to reinfection or not [122]. A single case of molecular proven reinfection by another *E. chaffeensis* strain has been documented in a liver transplant patient [140]. Of note, one case of a despite appropriate antimicrobial treatment (tetracycline and chloramphenicol) persistent and finally fatal infection with *E. chaffeensis* has been reported and suggests that *Ehrlichia*, similarly to other intracellular pathogens (like e.g. *Coxiella*) may, albeit rarely, cause chronic persistent infection [88].

## Limitations

Our analyses have several limitations. Next to publication bias and the retrospective nature of most available data sets, most reviewed studies did only provide incomplete data sets. In addition to the inhomogeneity and incompleteness of the available data sets, the data and results of studies that reported on case series or cohorts were often given as total values, medians or percentages, so that it was often not possible to assign the data to individual cases (see section on CRNID). Even though we tried to eliminate case duplicates, the cohort studies we looked at could potentially contain duplicate cases already described in other cohort studies or case reports. When we compare our findings in case studies with the cohorts we looked at as well as CDC data on HME in the USA [119], we see that the course of illness in our cases was comparatively severe, which points to reporting and publication bias. The latter particularly impacts systematic review being based on a considerable number of case reports, as severe cases are more likely to be reported and published than mild cases.

Box 1 summarizes the main conclusions we have drawn from our review and Box 2 contains our selection of publications that we recommend clinicians to read.

## Box 1

**Key learning points:**

- There is little evidence to support the existence of HME (*E. chaffeensis-*) endemic regions outside North America. The very few sporadic cases of human *Ehrlichia* infections reported from other regions of the world are based on low diagnostic certainty or report cases of human infections with other *Ehrlichia* species

- Although primarily tick-borne, transmission by solid organ transplantation and blood-transfusion is reported

- HME usually presents as a non-specific febrile illness accompanied by cytopenia (primarily thrombocytopenia, leukopenia) and elevated liver function tests

- Although usually mild and self-limiting, HME may cause severe and even life-threatening complications, particularly in immunocompromized patients

- Secondary hemophagocytic lymphohistiocytosis (sHLH) is an above-average complication of HME, but appears to have a more favorable outcome compared to sHLH of other causes

- The antimicrobial treatment of choice is with doxycycline and treatment response is usually fast with fever subsiding within 2 day after starting treatment

- The case fatality rate of HME is higher in immunocompromised compared to immunocompetent patients and sequelae are rare

## Box 2

**Key references:**

1. Dahlgren FS, Mandel EJ, Krebs JW, Massung RF, McQuiston JH. Increasing incidence of *Ehrlichia chaffeensis* and *Anaplasma phagocytophilum* in the United States, 2000–2007. Am J Trop Med Hyg. 2011;85(1):124–31. Epub 2011/07/08. doi: 10.4269/ajtmh.2011.10–0613 PubMed PMID: 21734137; PubMed Central PMCID: PMCPMC3122356.

2. Biggs HM, Behravesh CB, Bradley KK, Dahlgren FS, Drexler NA, Dumler JS, et al. Diagnosis and Management of Tickborne Rickettsial Diseases: Rocky Mountain Spotted Fever and Other Spotted Fever Group Rickettsioses, Ehrlichioses, and Anaplasmosis—United States. MMWR Recomm Rep. 2016;65(2):1–44. Epub 2016/05/14. doi: 10.15585/mmwr.rr6502a1 PubMed PMID: 27172113.

3. Ismail N, McBride JW. Tick-Borne Emerging Infections: Ehrlichiosis and Anaplasmosis. Clinics in Laboratory Medicine. 2017;37(2):317–40. doi: 10.1016/j.cll.2017.01.006

4. Otrock ZK, Eby CS, Burnham CAD. Human ehrlichiosis at a tertiary-care academic medical center: Clinical associations and outcomes of transplant patients and patients with hemophagocytic lymphohistiocytosis. Blood Cells, Molecules, and Diseases. 2019;77:17–22. doi: 10.1016/j.bcmd.2019.03.002

5. Kuriakose K, Pettit AC, Schmitz J, Moncayo A, Bloch KC. Assessment of Risk Factors and Outcomes of Severe Ehrlichiosis Infection. JAMA Netw Open. 2020;3(11):e2025577. Epub 2020/11/18. doi: 10.1001/jamanetworkopen.2020.25577. PubMed PMID: 33201233; PubMed Central PMCID: PMCPMC7672514.

## Supporting information

**S1 Text. Search terms used.**
(DOCX)

**S2 Text. Systematic review protocol.**
(DOCX)

**S3 Text. Laboratory reference values used.**
(DOCX)

**S4 Text. Data extraction sheet used for screening and selecting eligible publications.**
(DOCX)

**S5 Text. Reference list of considered publications.**
(DOCX)

**S6 Text. PRISMA checklist.**
(DOCX)

**S7 Text. Additional analyses of CRNID.**
(DOCX)

**S8 Text. Additional performed analyses of CRID with coinfections.**
(DOCX)

**S1 Table. Master table of raw data.**
(XLSX)

## Author Contributions

**Conceptualization:** Esther Kuenzli, Andreas Neumayr.

**Data curation:** Larissa Gygax, Sophie Schudel.

**Formal analysis:** Larissa Gygax, Christian Kositz.

**Methodology:** Christian Kositz, Esther Kuenzli, Andreas Neumayr.

**Resources:** Sophie Schudel.

**Supervision:** Christian Kositz, Esther Kuenzli, Andreas Neumayr.

**Visualization:** Larissa Gygax, Sophie Schudel.

**Writing – review & editing:** Larissa Gygax, Christian Kositz, Esther Kuenzli, Andreas Neumayr.

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
