## [Decision Letter · Decision Letter 0]

14 May 2024

Dear Dr. Kositz,

Thank you very much for submitting your manuscript "Human monocytotropic ehrlichiosis – a systematic review and analysis of the literature" for consideration at PLOS Neglected Tropical Diseases. As with all papers reviewed by the journal, your manuscript was reviewed by members of the editorial board and by two independent reviewers. In light of the reviews (below this email), we would like to invite the resubmission of a significantly-revised version that takes into account the reviewers' comments. 

We cannot make any decision about publication until we have seen the revised manuscript and your response to the reviewers' comments. Your revised manuscript is also likely to be sent to reviewers for further evaluation.

Sincerely,

Wen-Ping Guo

Academic Editor

Georgios Pappas

Section Editor

Reviewer's Responses to Questions

**Key Review Criteria Required for Acceptance?**

**Methods**

-Are the objectives of the study clearly articulated with a clear testable hypothesis stated?

-Is the study design appropriate to address the stated objectives?

-Is the population clearly described and appropriate for the hypothesis being tested?

-Is the sample size sufficient to ensure adequate power to address the hypothesis being tested?

-Were correct statistical analysis used to support conclusions?

-Are there concerns about ethical or regulatory requirements being met?

Reviewer #1: (No Response)

Reviewer #2: Yes

**Results**

-Does the analysis presented match the analysis plan?

-Are the results clearly and completely presented?

-Are the figures (Tables, Images) of sufficient quality for clarity?

Reviewer #1: (No Response)

Reviewer #2: Yes

**Conclusions**

-Are the conclusions supported by the data presented?

-Are the limitations of analysis clearly described?

-Do the authors discuss how these data can be helpful to advance our understanding of the topic under study?

-Is public health relevance addressed?

Reviewer #1: (No Response)

Reviewer #2: Yes

**Editorial and Data Presentation Modifications?**

Reviewer #1: (No Response)

Reviewer #2: Data presentation could be improved to emphasize their unique findings of this review.

**Summary and General Comments**

Reviewer #1: The manuscript "Human monocytotropic ehrlichiosis – a systematic review and analysis of the literature" reviewed the Ehrlichia and ehrlichiosis worldwide systematically. The review is meaningful and the workload is quite large. However, there are also numerous problems. My comments are listed below:

1. The Article type should be Review, not Research Article.

2. Line 70, "Ehrlichioses and anaplasmoses" should be "Ehrlichiosis and anaplasmosis"?

3. What does "The different Anaplasmataceae" mean? Do you mean different members of Anaplasmataceae?

4. The Introduction section lacks many references.

5. Lines 88-93, this sentence is hard to read. Please rephrase it.

6. Lines 99-102, this sentence is hard to read. Please rephrase it.

7. Lines 102-103, "of both, E. chaffeensis and E. ewingii," should be "of both E. chaffeensis and E. ewingii".

8. Lines 108 and 111, "15'527" and "100'000" shoud be "15, 527" and "100, 000". Please revise it throughout the manuscript.

9. Line 114, "which’s enzootic" seem should be "whose enzootic".

10. Table 1, Table 6, and Lines 636-644. I would not recommend you cite and describe "Candidatus Ehrlichia erythraense". This is not a formally recognized Ehrlichia species.

11. Lines 131 and 133, "human granulocytotropic ehrlichiosis" and "human granulocytotropic anaplasmosis" should not be italicized.

12. Line 135, "«ehrlichiosis»" . It is rare to use "«»".

13. Line 138, "[add reference later]"?? You forgot to add the reference?

14. Line 172, "Besides microscopy" should be "Except for microscopy"?

15. Table 4, Taiwan is a province of China. Please do not list it separately. Also, you can show it as Taiwan (China).

16. Table 14, The "Outcome", "cure" should be "cured"?

17. Line 607, 615, and elsewhere in the manuscript. Is "HE" the same as "HME"?

18. I think the manuscript need proofreading by English native speaker.

Reviewer #2: This manuscript reviewed the epidemiology, diagnostics, cases, clinical signs, and outcome of HME. Although most HME cases occurred in North America, more and more records of infection have been reported around the world. In addition, a novel pathogen, Candidatus Ehrlichia erythraense, has been identified recently. In this manuscript, the authors performed a thorough literature search and analyzed human ehrlichiosis from various aspects. The results not only provided us a comprehensive knowledge but some new insights into the clinical manifestations and complications of human ehrlichiosis.

There are a few points to consider to make the manuscript suitable for publication. 

1. In abstract, the authors described, “Cases of HME are exclusively reported from North America.” (p2 L38) “Human monocytotropic ehrlichiosis (HME) is a bacterial disease caused by Ehrlichia chaffeensis which is transmitted by tick bites and exclusively reported from Northern America.” (p2 L52-54) In fact, cases of HME have been reported in various countries on different continents (p15-16 Table 4,5 Fig 5). The word “exculsively” was confusing.

2. Table 1 was not mentioned in the context.

3. Captions for tables are placed above the table, and captions for figures are placed below the figure.

4. Although I appreciate the authors’ efforts, the Results are tediously long. Certain tables could be combined, e.g. Table 4 and 5; while some figures could be omitted or left in supporting files. In this way, interesting findings, such as rash which appeared much more often in children, and HLH which was associated with human ehrlichiosis at a surprisingly high rate, could be further addressed.

5. “…, the low rate of coinfection appears quite plausible when considering that most other tick-borne pathogens in HE endemic areas are transmitted by Ixodes spp. ticks and not by tick species transmitting HE.” Although human ehrlichiosis is transmitted primarily by the lone star tick (Amblyomma americanum) in north America, the blacklegged tick (Ixodes scapularis), which is known able to transmit Lyme disease and babesiosis, is also an important vector of human ehrlichiosis in the area. Moreover, Ehrlichia has been identified in Ixodes and other ticks, e.g. Ixodes ricinus, Ixodes pacificus, Dermacentor variabilis, and Ixodes persulcatus.

PLOS authors have the option to publish the peer review history of their article (what does this mean?). If published, this will include your full peer review and any attached files.

Reviewer #1: No

Reviewer #2: No
---

## [Decision Letter · Decision Letter 1]

17 Jul 2024

Dear Dr. Kositz,

We are pleased to inform you that your manuscript 'Human monocytotropic ehrlichiosis – a systematic review and analysis of the literature' has been provisionally accepted for publication in PLOS Neglected Tropical Diseases.

Best regards,

Wen-Ping Guo

Academic Editor

Georgios Pappas

Section Editor

Reviewer's Responses to Questions

**Key Review Criteria Required for Acceptance?**

**Methods**

-Are the objectives of the study clearly articulated with a clear testable hypothesis stated?

-Is the study design appropriate to address the stated objectives?

-Is the population clearly described and appropriate for the hypothesis being tested?

-Is the sample size sufficient to ensure adequate power to address the hypothesis being tested?

-Were correct statistical analysis used to support conclusions?

-Are there concerns about ethical or regulatory requirements being met?

Reviewer #1: (No Response)

**Results**

-Does the analysis presented match the analysis plan?

-Are the results clearly and completely presented?

-Are the figures (Tables, Images) of sufficient quality for clarity?

Reviewer #1: (No Response)

**Conclusions**

-Are the conclusions supported by the data presented?

-Are the limitations of analysis clearly described?

-Do the authors discuss how these data can be helpful to advance our understanding of the topic under study?

-Is public health relevance addressed?

Reviewer #1: (No Response)

**Editorial and Data Presentation Modifications?**

Reviewer #1: (No Response)

**Summary and General Comments**

Reviewer #1: (No Response)

PLOS authors have the option to publish the peer review history of their article (what does this mean?). If published, this will include your full peer review and any attached files.

Reviewer #1: No

---

## [Editor Report · Acceptance letter]

28 Jul 2024

Dear Dr. Kositz,

We are delighted to inform you that your manuscript, "Human monocytotropic ehrlichiosis – a systematic review and analysis of the literature," has been formally accepted for publication in PLOS Neglected Tropical Diseases.

Best regards,

Shaden Kamhawi

co-Editor-in-Chief

Paul Brindley

co-Editor-in-Chief
